# Developmental decrease of entorhinal-hippocampal communication in immune-challenged DISC1 knockdown mice

Xiaxia Xu [1✉], Lingzhen Song[1], Rebecca Kringel[1] & Ileana L. Hanganu-Opatz [1✉]

The prefrontal-hippocampal dysfunction that underlies cognitive deficits in mental disorders emerges during early development. The lateral entorhinal cortex (LEC) is tightly inter-connected with both prefrontal cortex (PFC) and hippocampus (HP), yet its contribution to the early dysfunction is fully unknown. Here we show that mice that mimic the dual genetic (G) -environmental (E) etiology (GE mice) of psychiatric risk have poor LEC-dependent recognition memory at pre-juvenile age and abnormal communication within LEC-HP-PFC networks throughout development. These functional and behavioral deficits relate to sparser projections from LEC to CA1 and decreased efficiency of axonal terminals to activate the hippocampal circuits in neonatal GE mice. In contrast, the direct entorhinal drive to PFC is not affected, yet the PFC is indirectly compromised, as target of the under-activated HP. Thus, the entorhinal-hippocampal circuit is already impaired from neonatal age on in GE mice.

[1] Institute of Developmental Neurophysiology, Center for Molecular Neurobiology, University Medical Center Hamburg-Eppendorf, 20251 Hamburg, Germany. ✉email: xiaxia.xu@zmnh.uni-hamburg.de; hangop@zmnh.uni-hamburg.de

The major burden of major psychiatric disorders, such as schizophrenia, is a lifelong cognitive disability[1,2]. Its devastating impact on the daily life is augmented by the fact that the available medication causes a weak, if any, improvement of cognitive deficits concerning attention, processing speed, working and long-term memory, executive function, and social cognition, despite (almost) complete resolution of psychotic symptoms[3]. The pathophysiological substrate of these deficits has been identified to center on the prefrontal-hippocampal network[4]. Abnormal prefrontal-hippocampal coupling during working memory tasks has been described in schizophrenia patients[4,5]. It relates not only to local alterations at microscopic and macroscopic scales in both areas[6] but also to connectivity dysfunction within large-scale networks[7]. In particular, the interactions between HP and entorhinal cortex (EC), which tightly communicate with each other along reciprocal pathways[8], have been addressed in clinical and neuropathological investigations[9,10]. While some findings are still controversial, cytoarchitectural disorganization, cellular and synaptic deficits in layer 2 as well as aberrant axonal innervation have been detected in the EC of schizophrenia patients[9,11,12]. It has been hypothesized that these deficits result from developmental disturbance of migration, differentiation and wiring of entorhinal circuits[13,14]. However, their underlying mechanisms and contribution to the cognitive impairment are currently unknown.

Broader network dysconnectivity as underlying mechanism of cognitive deficits[15,16] has been also identified in rodent models of psychiatric disorders[17,18]. While these models vary in their etiology, utility and validity, they provide precious insights into the neural basis of cognitive deficits that are difficult to obtain in humans[19], especially when considering areas poorly investigated and less accessible, such as LEC. It has been shown that LEC lesion causes disease-characteristic altered mesolimbic dopaminergic transmission[20,21]. Being a direct correlate of cognitive performance[22], gamma oscillations within entorhinal-hippocampal circuits are disrupted in mouse models of psychiatric illness[23].

Besides providing insights into the neurobiological substrate of disease, mouse models enable to test the neurodevelopmental origin of illness-related deficits. Recently, we showed that already during early postnatal development, the neuronal activity and communication between HP and PFC are profoundly compromised in a mouse model mimicking the combined genetic (knockdown of Disrupted-In-Schizophrenia 1 (*Disc1*)[24]) and environmental (maternal immune activation (MIA)) etiology (GE) of psychiatric risk[25–29], but largely normal in single-hit genetic (G, DISC1 alone) or environmental (E, MIA alone) model[25,26]. Weaker PFC-HP communication at neonatal age results from dysfunction of local microcircuit as well as sparsification of hippocampal projections targeting the PFC[27–30]. At this age, LEC boosts the prefrontal-hippocampal circuits by projecting to both PFC and HP[8,31] and facilitating their oscillatory entrainment[32]. However, an in-depth characterization of the LEC role for PFC-HP communication in disease models is currently lacking.

To address this knowledge gap, we combine in vivo electrophysiology and optogenetics with anatomical tracing and behavioral investigation to interrogate the developing entorhinal-hippocampal-prefrontal circuits in the GE mouse model. We show that in neonatal GE mice, sparser and less efficient projections from LEC cause weaker activation of HP. In turn, the HP fails to sufficiently entrain the PFC, yet the direct entorhinal projections to PFC are largely intact.

## Results

### Pre-juvenile GE mice have poorer recognition memory.
Poorer associative memory has been identified both in first episode and chronic schizophrenia patients[33]. This memory form requires LEC but not the medial entorhinal cortex (MEC) integrity[34,35]. The strong LEC input to HP provides non-spatial (contextual) information and facilitates the binding of information relating to objects, places, and contexts. To get first insights into the LEC function towards the end of development, we assessed the associative recognition memory in mice of pre-juvenile age (postnatal day (P) 17–18). At this age, the mice already have fully developed sensory and motor abilities required for processing of novelty[36]. Two tasks were used: (i) novel object preference (distinct objects) (NOPd) task that requires an object-object association in adult mice[34] and (ii) object-location preference (OLP) task that relies on object-location association. To confirm that LEC is involved in the two tasks, we firstly performed cFos staining. The detected post-task strong cFos expression in the LEC of CON mice (supplementary Fig. 1) was complemented by a second approach directly testing whether LEC was necessary for the tasks. We injected AAV9_CaMKII_hM4Di_EGFP in LEC of P1 CON mice. The neuronal activity in LEC was decreased by exposure to the DREADD agonist 21 (compound 21, C21) 45 mins before the NOPd task (P17) or OLP task (P18). After C21 injection, the mice were able to recognize the novelty in NOP task, but not in NOPd and OLP task (supplementary Fig. 2). These data show that LEC was not only involved in, but also necessary for these two tasks.

To test whether GE mice show poorer LEC-dependent associative recognition memory, we investigated CON and GE mice in NOPd and OLP tasks. During the familiarization trial of the NOPd task, all mice spent equal time investigating the two objects placed in the arena (Fig. 1a). During the test trial, all mice spent longer time interacting with the novel object than with the familiar one (paired-sample t-test; CON: $n = 20$, $5.25 \pm 0.31$ s vs. $2.13 \pm 0.21$ s, $p = 4.63e{-}12$, df = 19, $t = -15.15$; GE: $n = 14$, $3.81 \pm 0.42$ s vs. $1.93 \pm 0.24$ s, $p = 0.002$, df = 13, $t = -3.86$) (Fig. 1a). However, GE mice had a poorer object discrimination than CON mice ($0.30 \pm 0.07$ vs. $0.44 \pm 0.03$, one-way ANOVA, $F_{(1,32)} = 4.27$, $p = 0.047$). The behavioral impairment was absent in one-hit G and one-hit E mice (supplementary Fig. 3a). During the OLP task, the pre-juvenile mice had to associate object and location and distinguish, which object was placed in a new location (Fig. 1b). In the test trial, CON mice spent significantly longer time with the new location-object than with the old location-object ($n = 19$, $5.79 \pm 1.19$ s vs. $2.22 \pm 0.25$ s, $p = 0.009$, df = 18, $t = -2.94$, paired-sample t-test). GE mice spent comparable time with the two objects ($n = 16$, $2.98 \pm 0.34$ s vs. $3.38 \pm 0.32$ s, $p = 0.35$, df = 15, $t = -0.96$, paired-sample t-test). Correspondingly, the object discrimination was significantly lower in GE mice than in CON mice ($0.03 \pm 0.08$ vs. $0.38 \pm 0.05$, one-way ANOVA, $F_{(1, 33)} = 12.68$, $p = 0.0011$). One-hit G and one-hit E mice were also unable to recognize the object with a new location (supplementary Fig. 4b). These data reveal that, besides deficits in object, location and recency recognition that have been previously identified in GE mice[25], the LEC-dependent recognition abilities are also impaired in pre-juvenile GE mice.

Poor performance in NOPd and OLP tasks may result not only specifically from dysfunction of entorhinal networks but also from poorer motor abilities and/or enhanced anxiety. To exclude these confounding effects, we analyzed the velocity of mice during familiarization and test trials of NOPd and OLP tasks (supplementary Fig. 3). The motor abilities of both groups were comparable as reflected by the similar velocity in the arena. We also analyzed the exploratory behavior of mice in the open field (10 min) at P16. All animals spent most of the time in the outer circle of the arena close to the walls and the distance covered in the inner circle was comparable between the groups (CON:

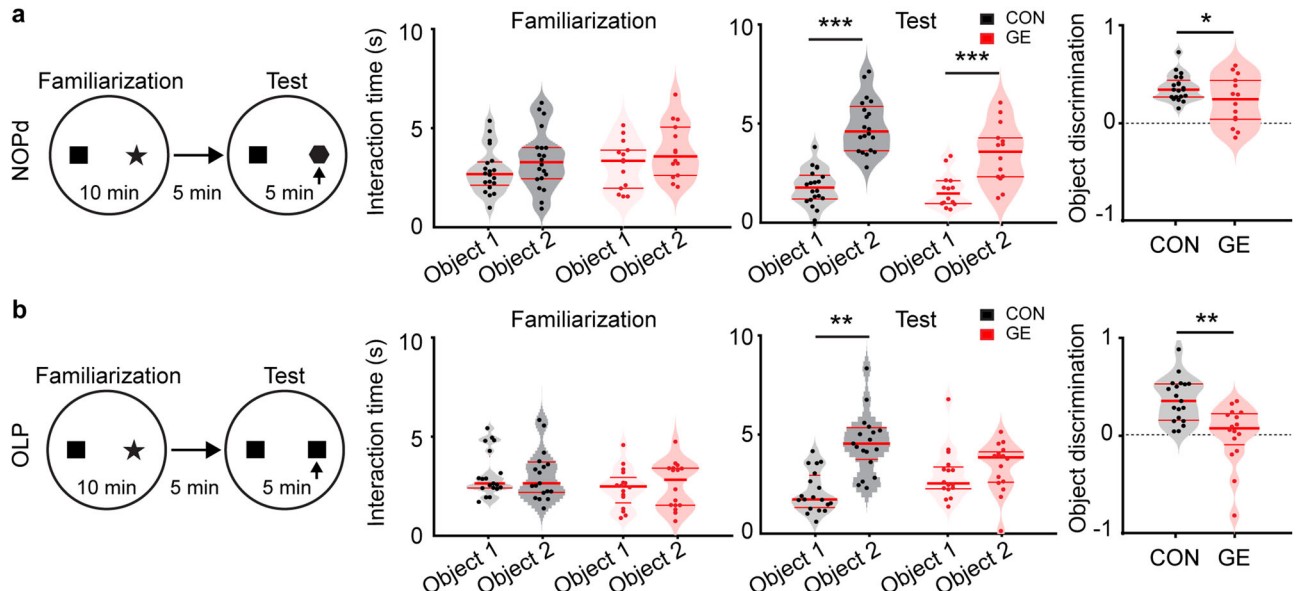

**Fig. 1 The performance of pre-juvenile GE mice in associative recognition memory tasks. a** Left, schematic of the protocol for NOPd task. Middle, violin plots displaying the interaction time (/min) spent by CON and GE with the objects during familiarization and test trials (Paired-sample $t$-test, $p = 4.63$e-12, df = 19, $t = -15.15$ for CON mice, $p = 0.002$, df =13, t = −3.86 for GE mice). Right, violin plots displaying the discrimination ratio in test trials (right) (one-way ANOVA, $F_{(1, 32)} = 4.27$, $p = 0.047$). **b** Same as **a** for OLP task (left, paired-sample $t$-test, $p = 0.009$, df = 18, $t = -2.94$ for CON mice, $p = 0.002$, $p = 0.35$, df = 15, $t = -0.96$ for GE mice; right, one-way ANOVA, $F_{(1, 33)} = 12.68$, $p = 0.0011$). In **a** and **b**, black dotted line indicates chance level. For violin plots, black and red dots correspond to individual animals and the red horizontal lines display the median as well as 25th and 75th percentiles. *$p < 0.05$, **$p < 0.01$, ***$p < 0.001$.

250 ± 48 cm; GE: 278 ± 51 cm). These data suggest that GE mice were not more anxious than CON mice.

Taken together, these results indicate that already at pre-juvenile age the LEC-dependent associative recognition memory is impaired in GE mice. This cognitive deficit complements the previously reported poor performance in other non-associative recognition tasks[25,27,28].

**The oscillatory entrainment of LEC within prefrontal-hippocampal networks is impaired in GE mice throughout development.** The above reported behavioral deficits might result from the developmental disruption of LEC function. To directly address this hypothesis, we investigated the oscillatory and firing activity of LEC and its embedding into PFC-HP networks. For this, we performed multisite extracellular recordings of local field potential (LFP) and multiple-unit activity (MUA) from LEC, simultaneously with the CA1 area of intermediate/ventral HP (i/vCA1) and prelimbic subdivision (PL) of PFC of urethane-anesthetized P20–23 CON, GE mice in vivo (Fig. 2). All investigated mice showed similar patterns of continuous network activity, which covered a broad frequencies spectrum and correspond to sleep-like rhythms mimicked by urethane anesthesia[37–39]. While the oscillatory power in LEC as well as PFC and HP was comparable in CON and GE mice (Fig. 2a), the firing activity (i.e. averaged single unit activity (SUA), CON: $n = 12$, GE: $n = 10$, Wilcoxon rank-sum test) was significantly increased in both entorhinal layer 5/6 (5.22 ± 0.94 vs. 2.62 ± 0.88, $p = 0.027$, zval = −2.20, ranksum = 104) and layer 2/3 (6.30 ± 0.86 vs. 3.18 ± 0.98, $p = 0.013$, zval = −2.47, ranksum = 100) as well as in stratum pyramidale of hippocampal CA1 area (5.79 ± 1.05 vs. 2.65 ± 0.69, $p = 0.019$, zval = −2.34, ranksum = 102) in GE mice (Fig. 2b). The prefrontal firing activity was also increased in GE mice, yet below significance threshold (layer 5/6: 6.65 ± 1.34 vs. 3.53 ± 1.49, $p = 0.07$, zval = −1.81, ranksum = 110; layer 2/3, 6.56 ± 1.31 vs. 4.11 ± 1.49, $p = 0.18$, zval = −1.35, ranksum = 117, Wilcoxon

rank-sum test). To assess the information flow within entorhinal-hippocampal-prefrontal networks, we used the generalized partial directed coherence (gPDC), a measure that reflects the directionality of network interactions in different frequency bands (Fig. 2c, Wilcoxon rank-sum test). The information flow of LEC- > PFC (0.079 ± 0.002 vs. 0.103 ± 0.013, $p = 0.008$, zval = 2.65, ranksum = 254) and HP- > PFC (0.077 ± 0.002 vs. 0.089 ± 0.005, $p = 0.005$, zval = 2.80, ranksum = 257) was significantly reduced in GE mice ($n = 11$) when compared to CON mice ($n = 15$). The information flow of LEC- > HP was comparable between the two groups (0.108 ± 0.005 vs. 0.115 ± 0.010, $p = 0.87$, zval = 0.016, ranksum = 206).

In line with previous studies[25,27,28], firing and coupling deficits at pre-juvenile age might reflect abnormal circuit wiring initiated at earlier stages of development. To test this hypothesis, we investigated the activity patterns within entorhinal-hippocampal-prefrontal networks in neonatal (P8-10) CON ($n = 14$) and GE ($n = 14$) mice (Fig. 3). Extracellular recordings of LFP and MUA showed that discontinuous spindle-shaped oscillations with frequency components peaking in theta band (4–12 Hz) inter-mixed with irregular low amplitude beta-gamma band components (12–30 Hz) were the dominant pattern of entorhinal network activity of both groups of mice (supplementary Fig. 5). The discontinuous oscillatory events classified as spindle-bursts were superimposed on a slow rhythm (2–4 Hz) that continuously entrained the neonatal LEC and related to respiration[40]. The occurrence and duration of discontinuous oscillatory events (4–30 Hz) were comparable in CON and GE mice (supplementary Fig. 6a). However, their power was significantly smaller ($p = 0.04$, $F_{(1,23)} = 4.69$, one-way ANOVA) in GE mice (4.81 ± 0.63) than CON mice (11.04 ± 2.60) (Fig. 3a). Given that single-hit E and G mice were indistinguishable in their activity patterns from CON mice (supplementary Fig. 6b), single-hit models were not further investigated. The diminished network activity in LEC was accompanied, as previously reported[25], by the

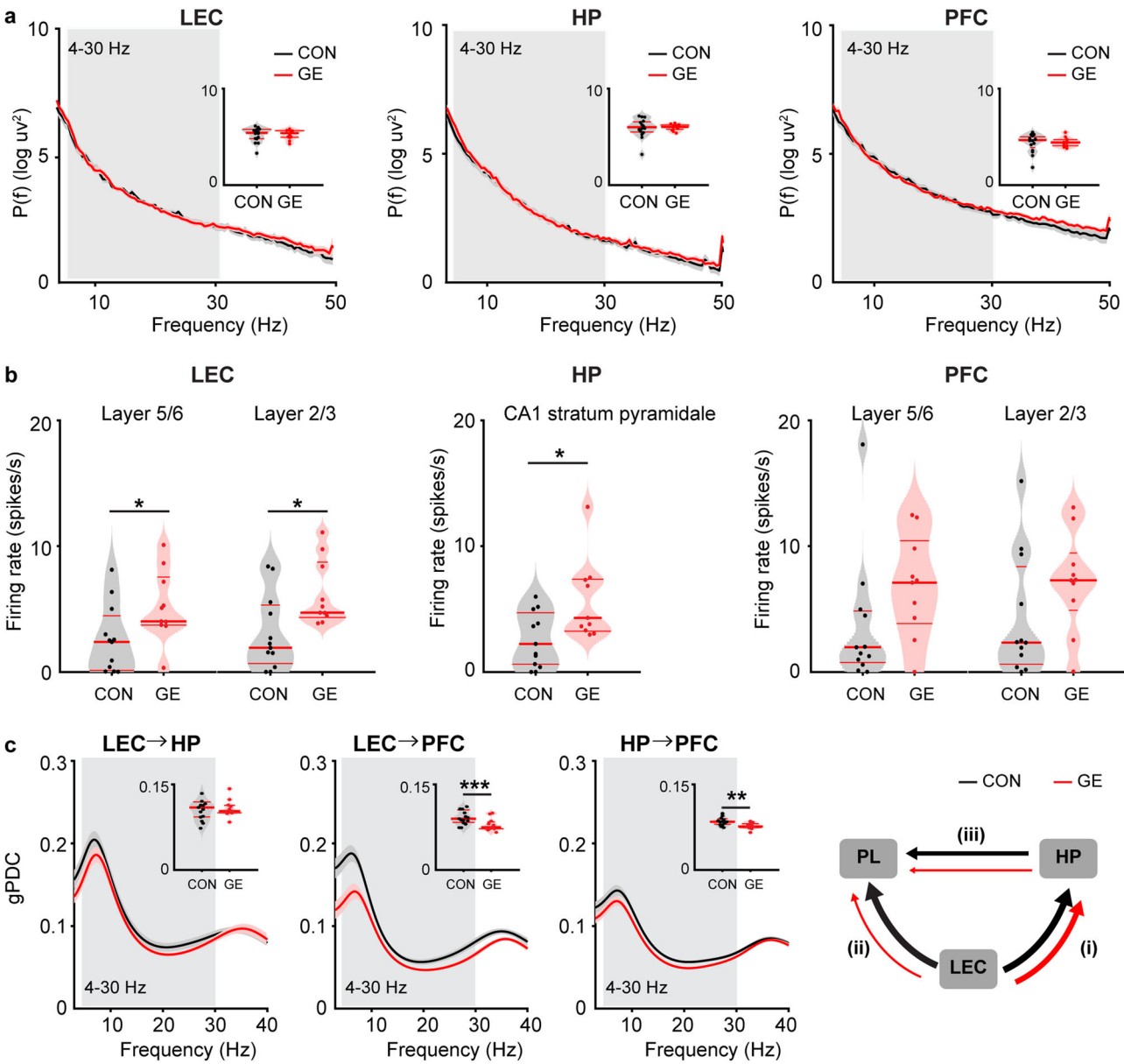

**Fig. 2 Patterns of network activity in LEC, HP, and PFC as well as functional communication within LEC-HP-PFC networks in pre-juvenile CON and GE mice. a** Averaged power spectra P(f) of oscillatory activity in CON and GE mice. Insets, violin plots displaying the power averaged for 1–30 Hz in CON and GE mice. **b** Violin plots displaying the firing rate in CON and GE mice (Wilcoxon rank-sum test, LEC layer 5/6: $p = 0.027$, zval $= -2.20$, ranksum $= 104$; LEC layer 2/3: $p = 0.013$, zval $= -2.47$, ranksum $= 100$; CA1, $p = 0.019$, zval $= -2.34$, ranksum $= 102$; PFC layer 5/6: $p = 0.07$, zval $= -1.81$, ranksum $= 110$; PFC layer 2/3: $p = 0.18$, zval $= -1.35$, ranksum $= 117$). **c** Line plots displaying the information flow measured by gPDC within LEC-HP-PFC circuits. Insets, violin plots displaying the gPDC when averaged for 4–30 Hz (Wilcoxon rank-sum test, LEC → PFC: $p = 0.008$, zval $= 2.65$, ranksum $= 254$; $p = 0.005$; HP → PFC: zval $= 2.80$, ranksum $= 257$; LEC → HP: $p = 0.87$, zval $= 0.016$, ranksum $= 206$). Right, schematic of information flow within LEC-HP-PFC circuits during pre-juvenile development as resolved by gPDC. The line thickness corresponds to the strength of information flow between brain regions. For violin plots, black and red dots correspond to individual animals and the red horizontal lines display the median as well as 25th and 75th percentiles. *$p < 0.05$, **$p < 0.01$, ***$p < 0.001$.

dysfunction of network activity in both HP and PFC (Fig. 3a). While the oscillatory power in LEC, PFC and HP was significantly reduced in GE mice, the firing activity (SUA) was comparable (Wilcoxon rank-sum test) in entorhinal layer 5/6 (0.53 ± 0.14 vs. 0.32 ± 0.08, $p = 0.54$, zval $= -0.62$, ranksum $= 189$), entorhinal layer 2/3 (0.37 ± 0.12 vs. 0.19 ± 0.07, $p = 0.30$, zval $= -1.03$, ranksum $= 180$), stratum pyramidale of hippocampal CA1 area (0.32 ± 0.10 vs. 0.45 ± 0.16, $p = 0.26$, zval $= 1.13$, ranksum $= 228$) as well as prefrontal layer 5/6 (0.11 ± 0.05 vs. 0.24 ± 0.08, $p = 0.18$, zval $= 1.36$, ranksum $= 233$) (Fig. 3b). In line with previous data[25], the firing activity in prefrontal layer 2/3

was significantly reduced in GE mice (0.17 ± 0.04 vs. 0.48 ± 0.14, $p = 0.04$, zval $= 1.98$, ranksum $= 246$).

Next, we questioned whether the dampening of oscillatory activity in LEC, HP and PFC during early development related to communication deficits between these areas. For this, we firstly assessed the coupling by synchrony between LEC and PFC-HP pathway in neonatal CON ($n = 14$) and GE ($n = 14$) mice by calculating the coherence of oscillatory events and considering only the imaginary part of it that was not corrupted by volume conductance[41]. A tight theta-beta band LEC-HP, HP-PFC coupling of spindle-bursts was detected in neonatal CON mice

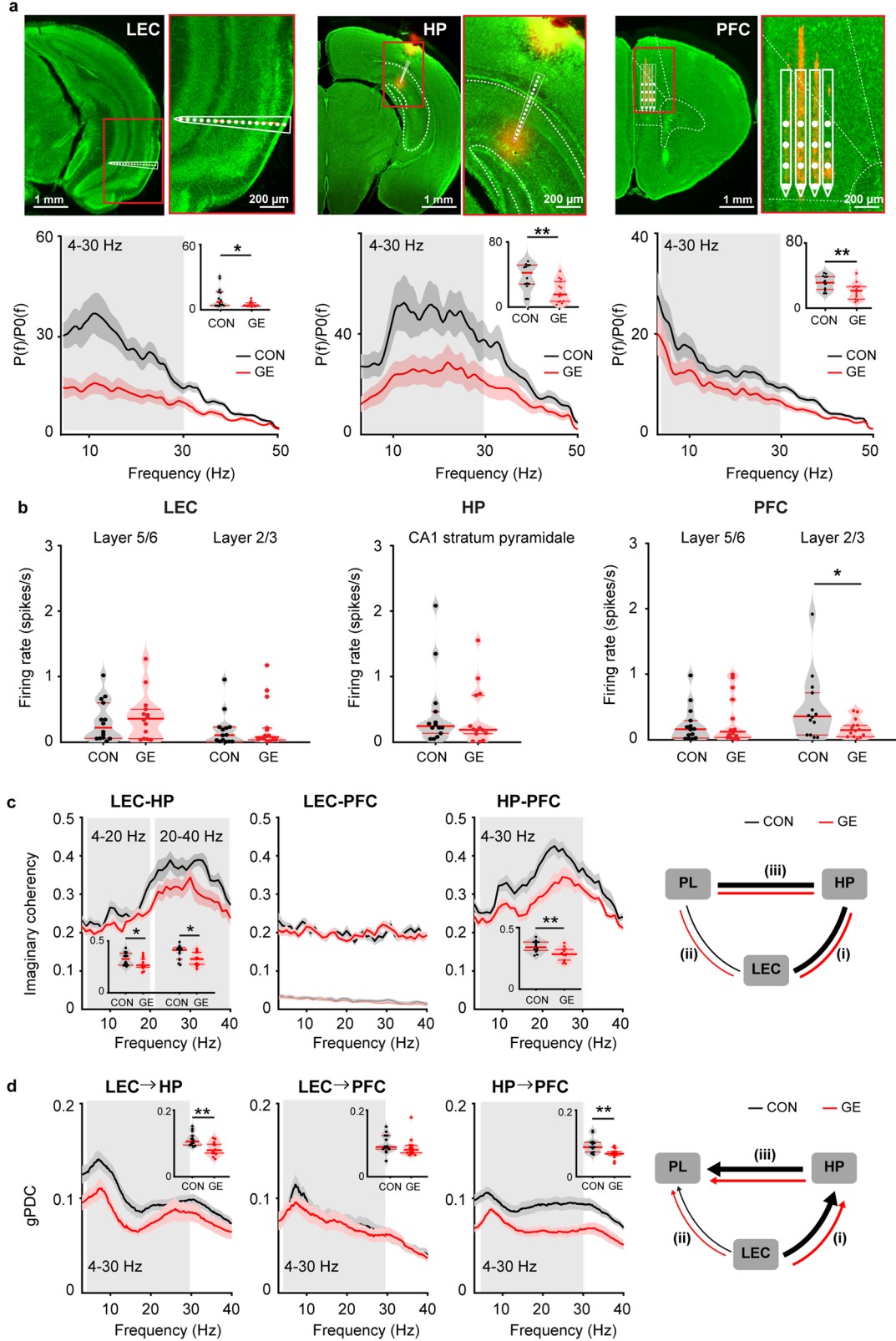

(Fig. 3c). In contrast, the imaginary coherence was significantly lower in GE mice (LEC-HP, 4–20 Hz: $0.286 \pm 0.016$ vs. $0.334 \pm 0.020$, $p = 0.04$, $F_{(1, 26)} = 4.45$; 20–40 Hz: $0.331 \pm 0.018$ vs. $0.397 \pm 0.019$, $p = 0.012$, $F_{(1, 26)} = 7.12$; HP-PFC: $0.345 \pm 0.013$ vs. $0.284 \pm 0.014$, $p = 0.002$, $F_{(1, 26)} = 11.340$, one-way ANOVA). The LEC-PFC coherence was much higher than the coherence calculated for the shuffled data in both CON and GE mice, yet no

frequency-specific coupling was detected. In a second step, gPDC was used to assess the directionality of information flow within entorhinal-hippocampal-prefrontal networks. In CON mice, we confirmed the previously reported drive HP- > PFC as well as the stronger information flow LEC → HP than LEC PFC[32,42]. The drive LEC→ HP was significantly decreased in GE mice ($0.084 \pm 0.006$ vs. $0.105 \pm 0.005$, $p = 0.008$, $F_{(1, 26)} = 8.39$, one-way

**Fig. 3 Patterns of network activity in LEC, HP, and PFC as well as functional communication within LEC-HP-PFC networks from neonatal CON and GE mice. a** Left, digital photomontage reconstructing the location of the DiI-labeled 1 × 16-site recording electrode (orange) in a 100 μm-thick coronal section containing the LEC from a P9 mouse, the position of recording sites (white dots) over LEC layers when displayed at higher magnification. Bottom, averaged power spectra P(f) of discontinuous oscillatory activity normalized to the baseline power P0(f) of time windows lacking oscillatory activity in CON ($n = 14$) and GE ($n = 14$) mice. Inset, violin plots displaying the average power spectra from 4–30 Hz in CON and GE mice (LEC: one-way ANOVA, $F_{(1, 23)} = 4.69$, $p = 0.04$). Middle, same as **a** for HP. Right, same as **a** for PFC. **b** Violin plots displaying the firing rate in CON and GE mice (Wilcoxon rank-sum test, LEC 5/6: $p = 0.54$, zval $= -0.62$, ranksum $= 189$; LEC 2/3: $p = 0.30$, zval $= -1.03$, ranksum $= 180$; CA1: $p = 0.26$, zval $= 1.13$, ranksum $= 228$; PFC 2/3: $p = 0.18$, zval $= 1.36$, ranksum $= 233$; PFC 5/6: $p = 0.04$, zval $= 1.98$, ranksum $= 246$). **c** Line plots of mean imaginary coherence for oscillatory activity simultaneously recorded in the LEC and HP, LEC, and PFC, as well as HP and PFC of CON (black) and GE (red) mice. The bottom lines in the coherence plots correspond to the imaginary coherence calculated from shuffled data. Insets for each coherence plot, violin plots displaying the imaginary coherence when averaged for 4–40 Hz (one-way ANOVA, LEC-HP: 4–20 Hz, $p = 0.04$, $F_{(1, 26)} = 4.45$; LEC-HP: 20–40 Hz, $p = 0.012$, $F_{(1, 26)} = 7.12$; HP-PFC: 4–30 Hz $p = 0.002$, $F_{(1, 26)} = 11.340$). Right, schematic of synchrony within LEC-HP-PFC networks during neonatal development as resolved by imaginary coherence. The line thickness corresponds to the coupling strength. **d** Same as **c** when the directional coupling within LEC-HP-PFC networks was estimated by gPDC (one-way ANOVA, LEC → HP: $p = 0.008$, $F_{(1, 26)} = 8.39$; HP → PFC: $p = 0.002$, $F_{(1, 26)} = 12.04$). For violin plots, black and red dots correspond to individual animals and the red horizontal lines display the median as well as 25th and 75th percentiles. *$p < 0.05$, **$p < 0.01$.

ANOVA), which, besides the previously reported local dysfunction in HP and PFC, might further contribute to the reduced drive HP→ PFC ($0.069 \pm 0.004$ vs. $0.094 \pm 0.006$, $p = 0.002$, $F_{(1, 26)} = 12.04$, one-way ANOVA) (Fig. 3d). The weak entorhinal drive to PFC was comparable in CON and GE mice.

Taken together, these results uncover the functional pathways of communication within neonatal limbic circuits with the LEC boosting the hippocampal activity in CON but not GE mice.

**Spatially distinct entorhinal projections to HP and PFC are sparser in neonatal GE mice.** One possible source of dysfunction within entorhinal-hippocampal-prefrontal circuits is the abnormal connectivity between these areas. We previously showed that already at the end of the first postnatal week hippocampal CA1 area strongly innervated the PFC, whereas no direct prefrontal projections targeted the HP. Moreover, we identified projections from LEC to HP as well as to PFC in neonatal rats[32]. However, it is unknown, whether the same or distinct entorhinal populations innervate PFC and HP. To elucidate the anatomical integration of LEC within the neonatal HP-PFC pathway, we injected the retrograde tracers CTB555 in HP and CTB488 in PFC of the same P7 mouse and monitored the projections after 3 days. CTB555 injection confined to i/vCA1 labeled cells mainly in layer 2/3 of LEC (Fig. 4a). CTB488 injection confined to prelimbic-infralimbic subdivisions of the PFC labeled cells in the same layers, yet in a distinct, more superficial part; labeled neurons were detected also in i/vCA1, confirming the previously described hippocampal projection to PFC. There was no overlap between the CTB555 and CTB488 labeled neurons in the superficial layers of LEC, indicating that the entorhinal HP- and PFC-projecting neurons had a distinct spatial organization. Moreover, assessment of histological identity of entorhinal neurons revealed that CA1-projecting neurons were calbindin-positive, whereas PFC-projecting neurons were reelin-positive (supplementary Fig. 7).

The substrate of decreased functional coupling within entorhinal-hippocampal-prefrontal networks in GE mice might be the sparser anatomical projections between the three areas. To test this hypothesis, we quantified the density of entorhinal HP- and PFC-projecting neurons in CON ($n = 6$) and GE mice ($n = 5$) (Fig. 4b, c). The density of HP-projecting neurons was higher than the density of PFC-projecting neurons in both CON and GE mice. GE mice showed a significantly ($p = 0.005$, $F_{(1, 9)} = 13.44$, one-way ANOVA) reduced density of HP-projecting neurons ($1556.14 \pm 132.40$) but a similar ($p = 0.87$, $F_{(1, 9)} = 0.03$, one-way ANOVA) density of PFC-projecting neurons ($784.26 \pm 94.51$) when compared with CON mice ($2156.84 \pm 123.05$, $761.99 \pm 105.82$).

Another source of dysfunction within entorhinal-hippocampal-prefrontal circuits in GE mice might represent the

LEC neurons *per se* that, due to abnormal properties, are not able to provide the activation relayed to downstream areas. To test this hypothesis, we performed in vitro whole-cell patch-clamp recordings from entorhinal neurons that were either retrogradely labeled by CTB488/Fluorogold (PFC-projecting neurons) or CTB555 (HP-projecting neurons). The passive membrane properties (RMP, $C_m$, $R_{in}$, Tm) of PFC- as well as HP-projecting neurons were similar in CON (27 neurons from 10 mice) and GE (12 neurons from 5 mice) mice (Table 1). All investigated neurons showed linear I–V relationships and their firing increased in response to depolarizing current injection. The active membrane properties (e.g., action potential (AP) threshold, AP amplitude, half-width, rheobase, firing frequency) of entorhinal neurons did not differ between CON and GE mice (Table 1). These results suggest that circuit dysfunction of GE mice does not mainly relate to cellular abnormalities of entorhinal PFC- and HP-projecting neurons.

**Weaker responsiveness of HP to optogenetic activation of LEC in neonatal GE mice.** To directly test the functional communication along axonal pathways within entorhinal-hippocampal-prefrontal networks, we monitored the responsiveness of the three areas to the activation of LEC. For this, we selectively transfected pyramidal neurons in LEC of CON ($n = 11$) and GE ($n = 13$) mice with a highly efficient fast-kinetics double mutant ChR2 (H134R) by micro-injections performed at P1 (Fig. 5a, supplementary Fig. 8, 9).

First, we assessed the firing probability induced by light stimulation in entorhinal pyramidal neurons in vivo. Blue light pulses (473 nm, 3 ms) at a frequency of 8 Hz led shortly (<10 ms) after the stimulus to precisely timed firing of transfected neurons in the LEC of both CON and GE mice (Fig. 5b). In line with previous investigations[43,44,45], the used light power did not cause local tissue heating that might interfere with cellular integrity, neuronal spiking, and network activity (supplementary Fig. 10). The light-induced firing probability was similar for both CON and GE groups (Fig. 5b). From the second pulse on, the firing of neurons from both groups gradually lost the precise timing to the stimulus and the response reliability, most likely due to the immaturity of entorhinal neurons unable to fire at the set frequency. Pulsed light stimulation in LEC led to rhythmic firing in HP, yet not in PFC (Fig. 5b). Quantification of the hippocampal light-induced firing probability revealed that CA1 neurons were reliably activated by light pulses in CON, yet not GE mice, most likely due to the reduced number of entorhinal HP-projecting neurons and the sparser entorhinal projections in HP (Fig. 4, supplementary Fig. 11).

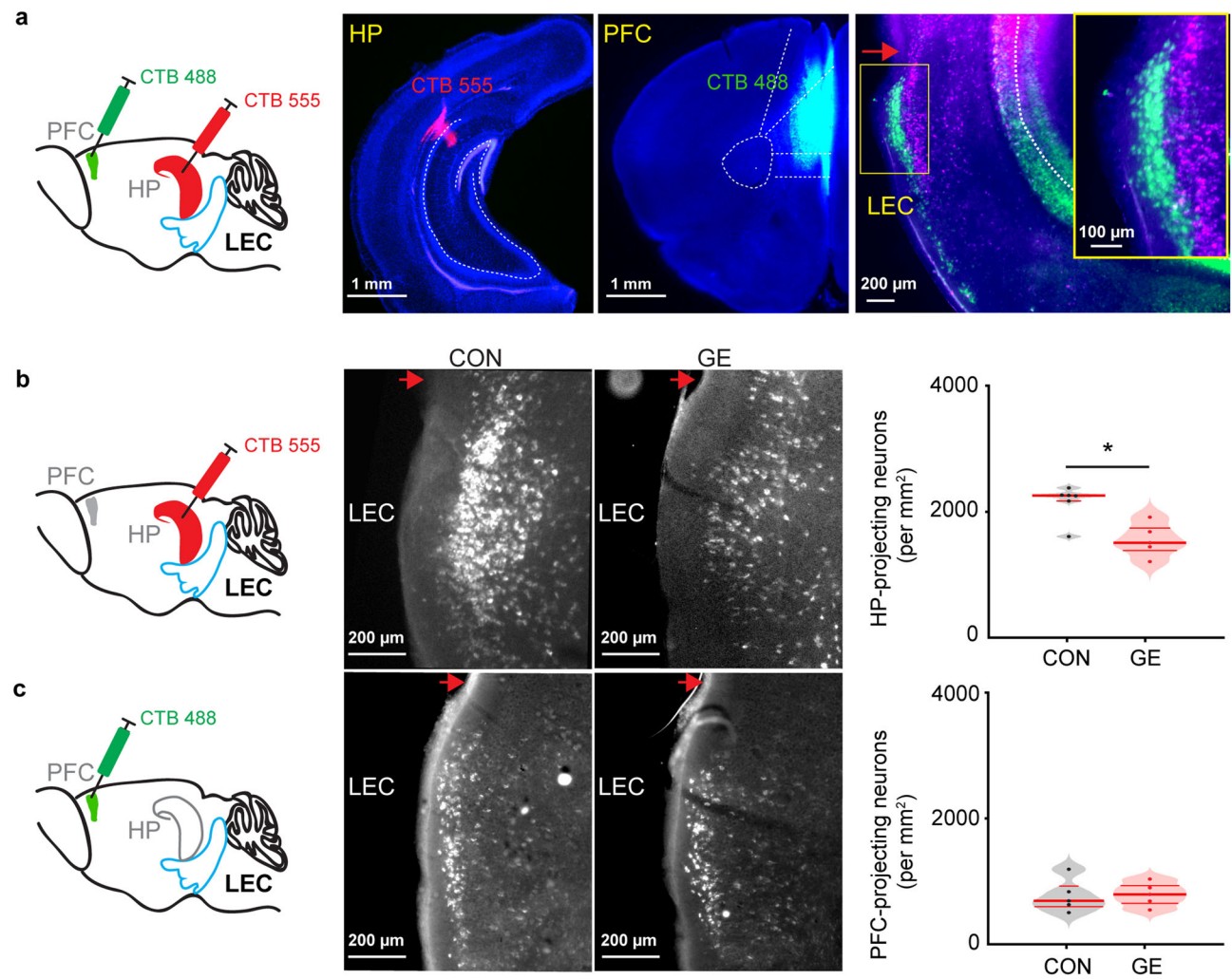

**Fig. 4 Long-range monosynaptic axonal projections connecting neonatal LEC, hippocampal CA1, and PFC. a** Schematic of the retrograde tracer CTB488 injection in PFC and CTB555 injection in HP. Digital photomontage showing the CTB555 injection in the HP (left) and CTB488 injection in the PFC (middle) of a P10 CON mouse (total $n = 3$ mice). Right, digital photomontage displaying CTB555- (red) and CTB488- (green) labeled neurons in LEC of the same mouse. Inset, labeled neurons in LEC shown at higher magnification. The red arrow indicates rhinal fissure. **b** Left, schematic of the retrograde tracer CTB555 injection in HP. Middle, photographs depicting CTB555-labeled neurons (white dot) in the LEC of a P10 CON and GE mouse. Right, violin plot displaying the number of CTB555-labeled neurons in the LEC of CON ($n = 6$) and GE ($n = 5$) mice (one-way ANOVA, $p = 0.005$, $F_{(1, 9)} = 13.44$). **c** Same as **b** for CTB488 injection in PFC (one-way ANOVA, $p = 0.87$, $F_{(1, 9)} = 0.03$). For violin plots, black and red dots correspond to individual animals and the red horizontal lines display the median as well as 25th and 75th percentiles. *$p < 0.05$.

**Table 1 Passive and active membrane properties of neurons in LEC of neonatal CON and GE mice.**

| | | HP-projecting neurons | | | PFC-projecting neurons | | |
|---|---|---|---|---|---|---|---|
| | | CON | GE | *p* | CON | GE | *p* |
| Passive properties | RMP (mV) | −68.76 ± 8.19 | −69.24 ± 7.44 | 0.12 | −69.19 ± 4.47 | −68.39 ± 5.47 | 0.99 |
| | $C_m$ (pF) | 111.85 ± 21.99 | 118.96 ± 14.10 | 0.46 | 120.32 ± 22.02 | 119.53 ± 14.05 | 0.96 |
| | $R_{in}$ (MΩ) | 606.99 ± 181.96 | 632.35 ± 102.39 | 0.62 | 481.20 ± 96.20 | 551.86 ± 93.68 | 0.27 |
| | $\tau_m$ (ms) | 67.24 ± 14.21 | 72.86 ± 16.08 | 0.17 | 58.95 ± 12.47 | 68.01 ± 12.95 | 0.29 |
| Active properties | AP threshold (mV) | −41.16 ± 7.06 | −42.20 ± 2.84 | 0.71 | −42.94 ± 2.61 | −44.05 ± 2.57 | 0.57 |
| | AP amplitude (mV) | 79.33 ± 5.29 | 77.88 ± 2.71 | 0.49 | 80.80 ± 3.38 | 81.21 ± 2.94 | 0.85 |
| | Half-width (ms) | 2.82 ± 0.45 | 2.61 ± 0.26 | 0.16 | 2.07 ± 0.31 | 1.92 ± 0.45 | 0.61 |
| | Rheobase (pA) | 45.61 ± 20.87 | 54.20 ± 16.42 | 0.69 | 56.07 ± 16.41 | 50.53 ± 8.27 | 0.49 |
| | Firing frequency (Hz) | 14.39 ± 4.53 | 14.79 ± 1.94 | 0.82 | 13.81 ± 3.05 | 16.67 ± 3.16 | 0.18 |

Second, to decide whether LEC activation boosts information flow within entorhinal-hippocampal-prefrontal networks, we investigated the synchrony of the three areas upon light stimulation. Ramp light stimulation (3 s, 473 nm) that enabled neurons to fire at their preferred and not a set frequency, augmented the 10-20 Hz LEC-HP coherence (0.14 ± 0.05) but not LEC-PFC coherence (−0.01 ± 0.02) in CON mice (Fig. 5c). Of note, frequency-specific boosting of HP through LEC activation

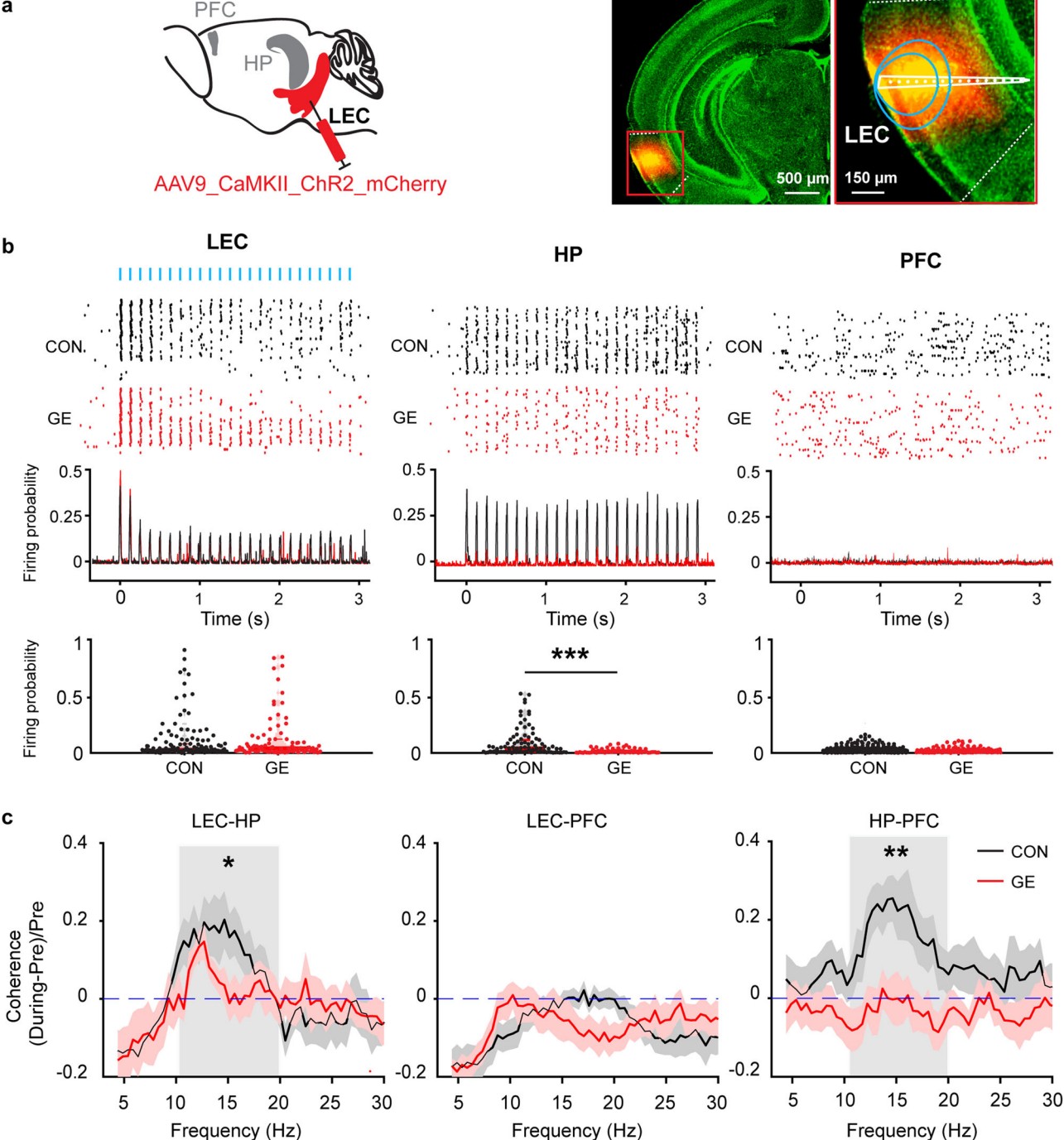

**Fig. 5 Light-induced activation of LEC. a** Schematic of AAV9-CaMKII-ChR2-mCherry injection in the LEC. Right, photographs depicting the injection position in the LEC of a P10 CON mouse (total $n = 11$ mice) and the position of injection site (red) shown at higher magnification. Blue lines correspond to the iso-contour lines for light power of 1 and 10 mW mm$^{-2}$, respectively. **b** Top, representative raster plot and corresponding spike probability histogram for LEC, HP and PFC in response to 50 sweeps of 8 Hz pulse stimulation (3 ms pulse length, 473 nm) in LEC. Bottom, violin plot displaying firing probability of single unit in LEC, HP, and PFC in response to 8 Hz light stimulation in LEC (Wilcoxon rank-sum test, HP: $p = 2.25e-6$, zval = 4.73, ranksum = 8324). **c** Line plots of coherence between LEC and HP, LEC, and PFC, and HP and PFC during ramp stimulation of LEC pyramidal neurons normalized to coherence values before stimulation (one-way ANOVA, LEC-HP: $p = 0.049$, $F_{(1,22)} = 4.10$). *$p < 0.05$, **$p < 0.01$, ***$p < 0.001$.

caused an indirect augmentation of HP-PFC synchrony in CON mice ($0.19 \pm 0.06$). In GE mice, the stimulation-induced LEC-HP coherence increase was of lower magnitude ($0.04 \pm 0.03$ vs. $0.14 \pm 0.05$, $p = 0.049$, $F_{(1,22)} = 4.10$) and consequently, not sufficient to augment HP-PFC synchrony ($-0.03 \pm 0.04$).

Taken together, these results indicate that LEC has a critical role for the activation of HP that on its turn boosts the entrainment of PFC. In contrast, the direct impact of entorhinal activity on PFC is low, if any. The LEC-driven activation of HP in GE mice is much weaker, being not further relayed to PFC.

**The function of entorhinal projections targeting the HP and PFC is selectively compromised in neonatal GE mice**. To experimentally backup the results above, we monitored the function of entorhinal projections targeting either the PFC or the HP. In all investigated P8-10 mice ($n = 24$), LEC projected to the prelimbic and infralimbic subdivisions of PFC, where it mainly targeted layer 5/6 neurons (Fig. 6a). Light stimulation (3 ms, 473 nm, 8 Hz) of terminals (Fig. 6b) in PFC of ChR2-transfected entorhinal neurons was performed simultaneously with extracellular LFP and MUA recordings in layer 5/6 of PL. Activation of entorhinal terminals caused a pronounced short-delay LFP response with a post-stimulus peak at 15 ms and augmented the neuronal firing in PFC of both CON and GE mice. The light-induced firing probability in layer 5/6 was low and comparable for CON and GE mice ($0.010 \pm 0.001$ vs. $0.009 \pm 0.001$, $p = 0.41$, zval $= -2.045$, ranksum$=15917$, Wilcoxon rank-sum test) (Fig. 6c). Only 15 out 138 (~10.87%) prefrontal neurons in CON mice and 11 out 108 (~10.19%) in GE mice increased their firing upon stimulation ($p = 0.97$, chi $= 0.013$, Chi-square test) (Fig. 6e, f).

To investigate the entorhinal projections to HP, we injected both Fluorogold and WGA tracers in the LEC of CON mice (Fig. 7a). The presence of WGA-labeled hippocampal neurons indicate that already at neonatal age, entorhinal projections target the HP. In line with previous studies[46], these projections accumulate in stratum lacunosum-moleculare (SLM) of CA1 area (Fig. 7b). The density of these projections significantly differed between CON and GE mice (supplementary Fig. 11). To test the function of entorhinal innervation of HP and whether the sparser projections in GE mice caused the network and neuronal deficits described above, we firstly performed patch-clamp recordings from visually-identified CA1 pyramidal neurons during light stimulation of entorhinal terminals in SLM in vitro (supplementary Fig. 12). In coronal slices from P9-10 mice, blue light pulses evoked excitatory postsynaptic currents (eEPSCs). The eEPSCs had a short latency (<9 ms) from stimulus onset and a fast kinetics. They were fully abolished by ionotropic AMPA receptor antagonists NBQX (10 μM) and NMDA receptor antagonists AP5 (50 μM) added to the bath solution. In GE mice, less neurons responded to light stimulation than in CON mice (9/42 vs. 16/26, $p = 0.0021$, chi $= 9.45$, Chi-square test). Moreover, GE mice had smaller eEPSC amplitude than CON mice ($19.59 \pm 9.04$ vs. $45.62 \pm 7.11$ pA, $p = 0.0042$, zval $= 2.86$, ranksum $= 259$, Wilcoxon rank-sum test) and showed a higher degree of variability upon stimulation as mirrored by the larger coefficient of variation ($0.52 \pm 0.073$ vs. $0.22 \pm 0.052$, $p = 0.0016$, $F_{(1, 23)} = 12.81$, One-way ANOVA). The kinetics of eEPSCs from GE mice was also disrupted, the events having a significantly longer rise-time decay when compared with those from CON mice ($4.65 \pm 0.23$ vs. $5.51 \pm 0.36$ ms, $p = 0.036$, $F_{(1, 23)} = 4.98$, One-way ANOVA). These results suggest that the entorhinal inputs on CA1 neurons are less efficient in GE mice.

Second, we performed multisite recordings of LFP and MUA in CA1 area during pulsed and ramp light stimulation of entorhinal terminals in HP (Fig. 7). The field response evoked by light pulses in HP had a fast (~15 ms) latency in all investigated mice, yet a smaller amplitude ($23.37 \pm 3.91$ μV, $p = 0.018$, zval $= 2.36$, ranksum $= 155$, Wilcoxon rank-sum test) in GE mice than CON mice ($51.49 \pm 13.06$ μV) (Fig. 7c). Current source density analysis (CSD) of light-evoked LFP response revealed the sink in stratum lacunosum of CA1 area (i.e. ~150–200 μm below stratum pyramidale) (Fig. 7d). The light-induced hippocampal firing probability was significantly ($p = 2.88\mathrm{e}10\text{-}7$, zval $= 5.13$, ranksum $= 7184$, Wilcoxon rank-sum test) lower in GE mice ($0.013 \pm 0.003$) than in CON mice ($0.072 \pm 0.019$) (Fig. 7e). This is in line with the lower ($p = 0.02$, chi $= 5.43$, Chi-square test) number of responsive hippocampal units in GE mice (8 out 73,

~11%) than in CON mice (22 out of 81, ~27%) (Fig. 7f, g). These results indicate that the function of entorhinal projections in HP is impaired in GE mice, their efficiency to boost the hippocampal activity being decreased.

If the function of entorhinal projections to PFC but not to HP is largely intact in GE mice, the question arises, whether the weaker entorhinal drive to HP is still sufficient to entrain the neural activity in PFC. Light activation of entorhinal terminals in HP (Fig. 8a) led to an increase of neuronal firing both in layer 5/6 ($1.33 \pm 0.10$) and layer 2/3 ($1.16 \pm 0.09$) of PFC in CON mice (Fig. 8b, c). In contrast, the stimulation has a significantly weaker, if any, effect in the PFC of GE mice (layer 5/6, $0.94 \pm 0.06$, $p = 0.0010$, zval $= 3.28$, ranksum $= 250$; layer 2/3: $0.94 \pm 0.06$, $p = 0.04$, zval $= 2.083$, ranksum$=221$; Wilcoxon rank-sum test). Correspondingly, the prefrontal-hippocampal coupling augmented in CON mice during stimulation ($0.09 \pm 0.03$), yet not in GE mice ($-0.02 \pm 0.004$, $p = 0.03$, $F_{(1, 25)} = 5.18$, one-way ANOVA, Fig. 8d).

To directly assess the entorhinal role within neonatal limbic circuits, we tested whether the decreased drive LEC- > HP in GE mice can be replicated by temporally precise inhibition of entorhinal terminals in HP in CON mice (Fig. 9). For this, we capitalized on the recently developed tool, the targeting-enhanced mosquito homolog of the vertebrate encephalopsin (eOPN3) that has been developed to selectively suppress neurotransmitter release at presynaptic terminals through the Gi/o signaling pathway[47]. Pulsed (473 nm, 5 ms, 8 Hz) light stimulation of eOPN3-expressing LEC terminals in HP of CON mice ($n = 13$) significantly (paired-sample t-test; $p = 0.008$, df $= 12$, t $= 3.17$) reduced the HP power ($-0.36 \pm 0.09$) that reached values comparable ($p = 0.158$, $F_{(1, 24)} = 2.12$, One-way ANOVA) to the hippocampal power reduction in GE mice when compared with CON mice ($-0.20 \pm 0.08$). Moreover, the selective silencing of LEC terminals in HP caused power decrease in the PFC ($-0.16 \pm 0.04$) that was significantly ($p = 0.016$, $F_{(1, 24)} = 6.75$, One-way ANOVA) smaller than the prefrontal power reduction in GE mice ($-0.41 \pm 0.09$). This result suggests that a disrupted entorhinal-hippocampal communication is one, but not the unique cause of abnormal prefrontal activity in GE mice. Correspondingly, the PFC-HP coherence was reduced in CON mice during silencing of entorhinal terminals in HP ($-0.23 \pm 0.03$), reaching comparable ($p = 0.79$, zval $= -0.27$, ranksum $= 176$) values to those calculated for GE mice ($-0.24 \pm 0.07$) (Fig. 9d).

Taken together, these results uncover that, while the sparse entorhinal projections to PFC seem to be structurally and functionally normal in GE mice, the entorhinal-hippocampal communication is impaired and has indirect effects on the prefrontal activity.

## Discussion
Mouse models mimicking the dual etiology of psychiatric risk, such as abnormal DISC1 function and immune challenge early in life, reproduce the neuronal network and, to a certain amount, behavioral deficits reported for patients. These deficits have been hypothesized to originate during development. Indeed, mice of an age corresponding to second-third gestational trimester in humans show prefrontal-hippocampal dysfunction[25,27–29], supporting the hypothesis of developmental miswiring in schizophrenia. The present results uncover novel mechanisms of miswiring in the neonatal brain and highlight the critical role of the LEC for the function of prefrontal-hippocampal circuits. We show that in GE mice (i) the patterns of oscillatory activity and coupling within LEC-HP-PFC networks are disrupted already at neonatal age, the entorhinal drive on HP being particularly

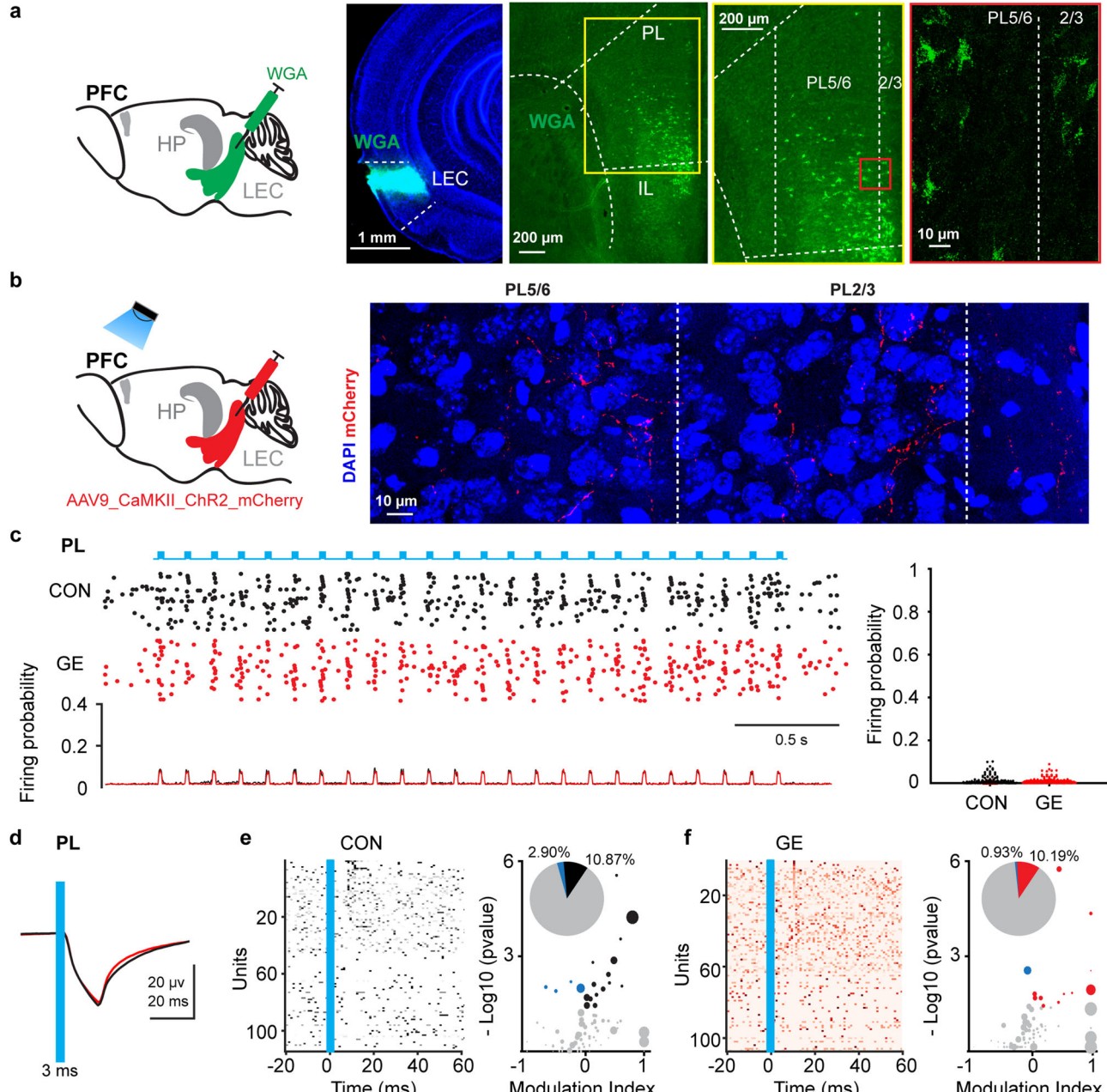

**Fig. 6 Prelimbic neurons innervated by LEC and light-induced activation of LEC terminals in PFC. a** Schematic of the anterograde trans-synaptic WGA injection in LEC. Photograph depicting the injection position in LEC of a P10 CON mouse (total $n = 4$ mice). Right, photographs depicting WGA-expressing neurons (green dots) in the PFC of a P10 CON mouse. WGA-labeled neurons in prelimbic layers displayed at a higher magnification. PL prelimbic subdivision of the PFC, IL infralimbic subdivision of the PFC. **b** Schematic of the AAV9-CaMKII-ChR2-mCherry injection in the LEC and light stimulation of entorhinal axonal terminals in PFC. Right, photograph depicting mCherry-labeled axons in PL5/6 and PL2/3 of a P10 CON mouse (total $n = 11$ mice). **c** Left, representative raster plot and corresponding spike probability histogram for prefrontal neurons in response to 50 sweeps of 8 Hz light stimulation. Right, violin plot displaying the firing probability of single units in LEC, HP and PFC in response to 8 Hz light stimulation in LEC (Wilcoxon rank-sum test, $p = 0.41$, zval $= -2.045$, ranksum $= 15917$). **d** Averaged LFP traces recorded in the prelimbic layer 5/6 in response to light stimulation of LEC terminals in CON (black) and GE (red) mice. The blue line indicates the 3 ms-long pulse stimulation in PFC. **e** Left, plot depicting the averaged firing of single cells in response to the first pulse stimulation from each sweep in CON mice. The blue line corresponds to the 3 ms-long pulse stimulation in PFC. Right, bubble plot depicting the modulation index of spiking response of prefrontal single units to pulse stimulation. Modulation index > 0 indicates increased firing activity, whereas values < 0 corresponds to decreased firing activity. The dot size mirrors the firing rate of single units. Inset, pie plot depicting the percentage of activated (black), inhibited (blue), and not changed (gray) prefrontal units upon light stimulation. **f** Same as **e** for GE mice.

reduced, (ii) the substrate of abnormal LEC-HP coupling is sparser entorhinal projections and their poorer efficiency to excite the HP, (iii) the direct entorhinal drive to PFC is largely unaffected, and (iv) the LEC-dependent recognition memory emerging at pre-juvenile age is impaired.

The EC represents a part of medial temporal lobe that is highly interconnected with the hippocampus and subcortical areas, such as the amygdala[48]. Related to its function and input-output connectivity, the EC has been divided into MEC that is involved in spatial navigation and spatial memory ("Where")[49] and LEC

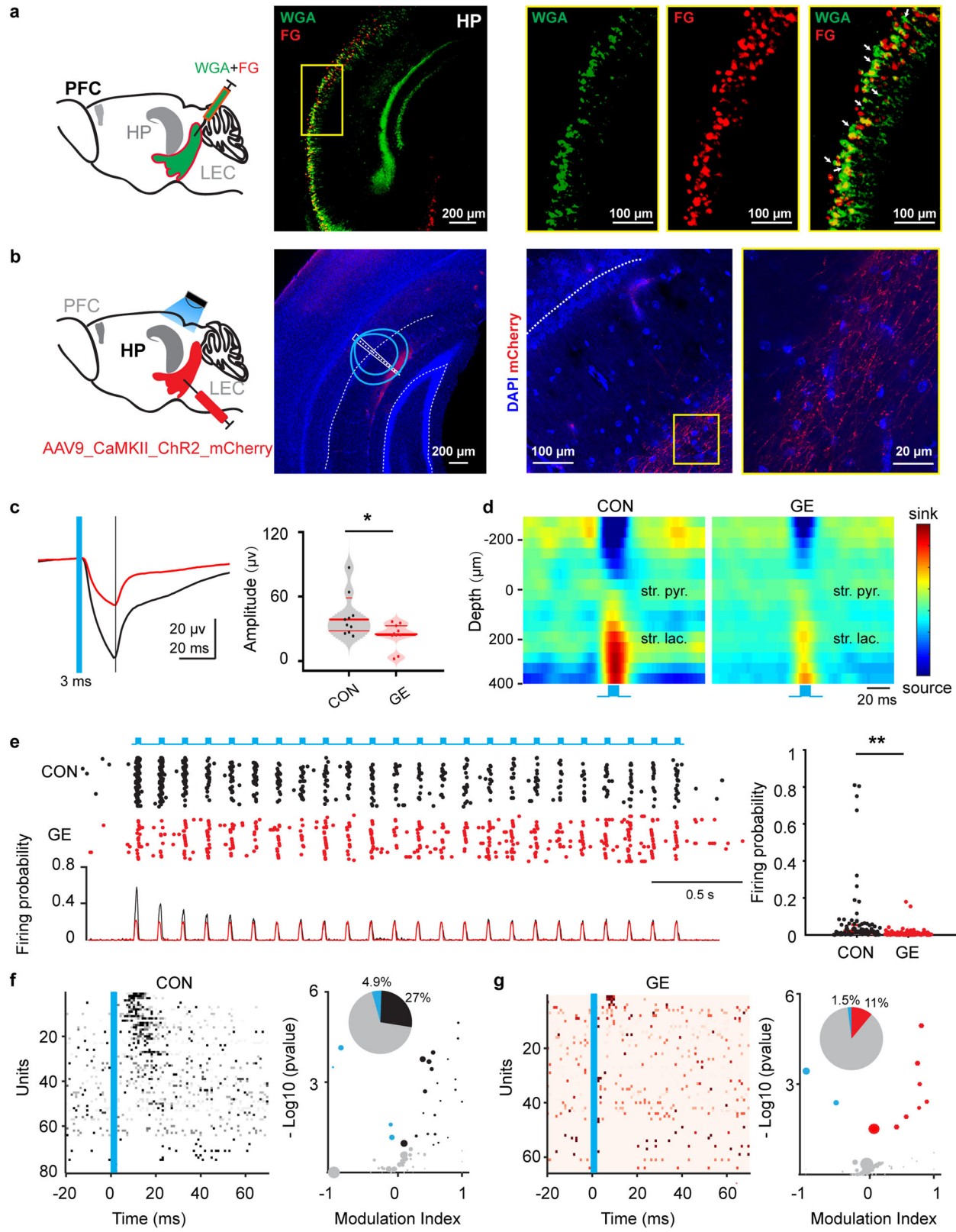

that codes context ("What") and temporal ("When") information[50,51]. In contrast to MEC, the LEC function has been less well dissected. It codes for object features and context-related locations, being critical for the performance in associative recognition memory of adults[34]. LEC is also involved in olfactory processing, as witnessed by lesions studies of LEC that led to olfactory anterograde amnesia[52] but also facilitation of olfactory

recognition[53]. Information transfer through LEC-HP synchrony is critical for olfactory associative learning[54]. Given the LEC function and its tight embedding into large-scale circuits, it is not surprising that LEC came into the focus when investigating the pathophysiology of major psychiatric illnesses. Clinical, post-mortem and imaging studies identified structural and synaptic deficits in LEC of schizophrenia and bipolar disorder

**Fig. 7 Light-induced activation of LEC terminals in HP. a** Left, schematic of the mixed Fluorogold (FG)and WGA injection in LEC. Middle, photographs depicting the neurons labeled by FG and WGA in the HP of a P10 CON mouse (total $n = 4$ mice). Right, photographs of the region marked by the yellow box when displayed at a higher magnification. **b** Left, schematic of the AAV9-CaMKII-ChR2-mCherry injection in LEC and light stimulation of entorhinal terminals in HP. Middle, photographs depicting the LEC axons labeled by mCherry (red) in the HP of a P9 CON mouse (total $n = 11$ mice). Blue lines correspond to the iso-contour lines for light power of 1 and 10 mW mm$^{-2}$, respectively. Right, mCherry-labeled axons in stratum lacunosum of CA1 displayed at a higher magnification. **c** Left, averaged hippocampal LFP in response to light stimulation of LEC terminals in CON and GE mice. The blue line indicates the pulse stimulation. Right, violin plots displaying the amplitude of the biggest response of the averaged LFP in HP (Wilcoxon rank-sum test, $p = 0.018$, zval = 2.36, ranksum = 155). **d** Representative LFP sinks and sources in response to 50 sweeps pulsed light stimulation of LEC terminals in the HP from a P9 CON mouse and a P9 GE mouse. **e** Left, representative raster plot and corresponding spike probability histogram for hippocampal neurons in response to 50 sweeps of 8 Hz light stimulation. Right, violin plot displaying the firing probability in HP in response to 8 Hz light stimulation (Wilcoxon rank-sum test, $p = 2.88e10-7$, zval = 5.13, ranksum = 7184). **f** Left, plot depicting the averaged firing of single hippocampal cells in response to the first pulse stimulation from each sweep in CON group. Right, bubble plot depicting the modulation index of spiking response of hippocampal single units to pulse stimulation. Modulation index > 0 indicates increased firing activity, whereas values < 0 correspond to decreased firing activity. The dot size mirrors the firing rate of single units. Inset, pie plot depicting the percentage of activated (black), inhibited (blue), and not changed (gray) prefrontal units upon light stimulation. **g** Same as **f** for GE mice. *$p < 0.05$, ***$p < 0.001$.

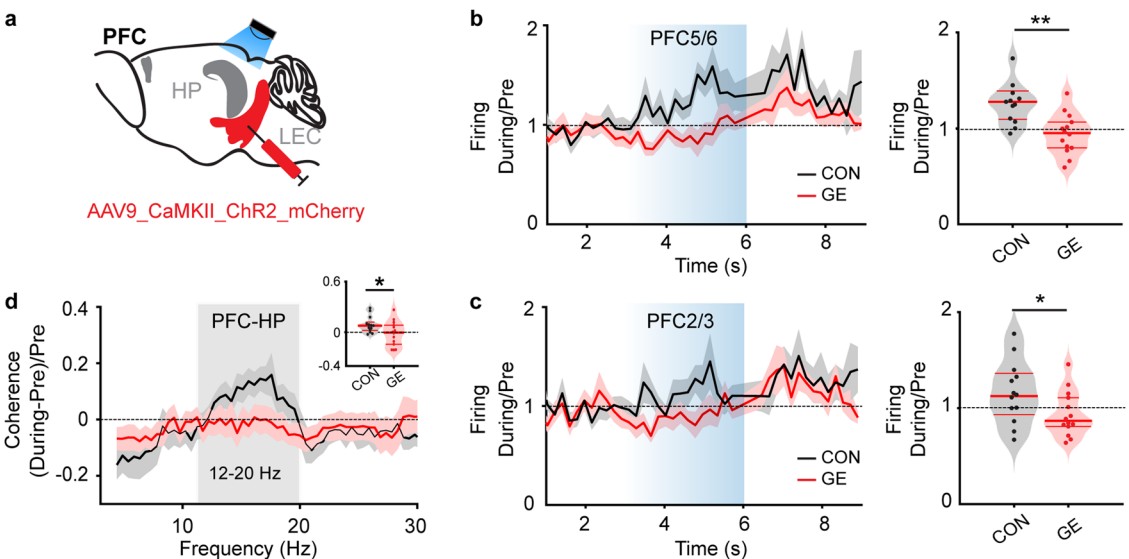

**Fig. 8 Firing activity in PFC during light-induced activation of LEC terminals in HP. a** Schematic of AAV9-CaMKII-ChR2-mCherry injection in LEC and light stimulation of entorhinal axonal terminals in HP. **b** Line plot of firing activity of prelimbic layer 5/6 neurons during 3 s-long ramp stimulation (light blue shadow) of LEC terminals in HP normalized to the activity before stimulation for CON and GE mice. The horizontal dotted line corresponds to no changes of firing activity during the stimulation. Right, violin plots displaying the firing activity of prelimbic layer 5/6 neurons during 3 s-long ramp stimulation normalized to the activity before stimulation (Wilcoxon rank-sum test, $p = 0.0010$, zval = 3.28, ranksum = 250). **c** Same as **b** for prelimbic layer 2/3 neurons (Wilcoxon rank-sum test, $p = 0.04$, zval = 2.083, ranksum = 221). (**d**) Line plots of HP-PFC coherence calculated during ramp stimulation of entorhinal terminals in HP and normalized to coherence values before stimulation. Inset, violin plot displaying the averaged 12–20 Hz coherence between HP and PFC during stimulation when normalized to coherence values before stimulation (one-way ANOVA, $p = 0.03$, $F_{(1, 25)} = 5.18$). For violin plots, black and red dots correspond to individual animals and the red horizontal lines display the median as well as 25th and 75th percentiles. *$p < 0.05$, **$p < 0.01$.

patients[55–57]. However, the mechanisms of these deficits remain largely unknown. It has been hypothesized that they result from abnormal development of LEC.

The present study addresses this hypothesis and uncovers the functional deficits of LEC-HP circuits and related behavior during development. Under physiological conditions, the neonatal LEC facilitates the hippocampal activation. In line with the density of projections, the direct entorhinal drive to HP is stronger than to PFC. The LEC-PFC coupling might occur via 3 distinct pathways: (i) monosynaptic connection from LEC to PFC, (ii) bi-synaptic transmission through direct synaptic connection from LEC to HP neurons, which further directly project to PFC, and (iii) poly-synaptic transmission through direct synaptic connection from LEC to HP neurons, which do not directly further project to PFC but through interplay with other hippocampal neurons. The anatomical investigation of projections and the double-tracing with WGA and CTB (supplementary Fig. 13) showed that the

pathways (i) and (ii), which would lead to high LEC-PFC coherence, are very weak, whereas the most prominent pathway (iii) mirrors the low LEC-PFC coherence. The prominent indirect and weak direct pathways together ensure the necessary level of neonatal prefrontal excitation and oscillatory activation, which are mandatory for adult prefrontal-related behavior[45].

The present study might also be instrumental for answering the question how non-sensory cortices, such as PFC, generate early oscillatory activity. Spontaneous activity from the periphery travels along axonal projections via brainstem and thalamic nuclei and boosts the entrainment of developing visual, barrel or auditory cortices in oscillatory rhythms that facilitate the emergence of characteristic functional topographies[58,59]. Such mechanisms are irrelevant for early prefrontal oscillations; here, it seems that LEC drives the activation patterns. This mechanism is not fully decoupled from sensory inputs, since LEC receives direct inputs from the olfactory bulb (OB). The blind and deaf mouse pups that

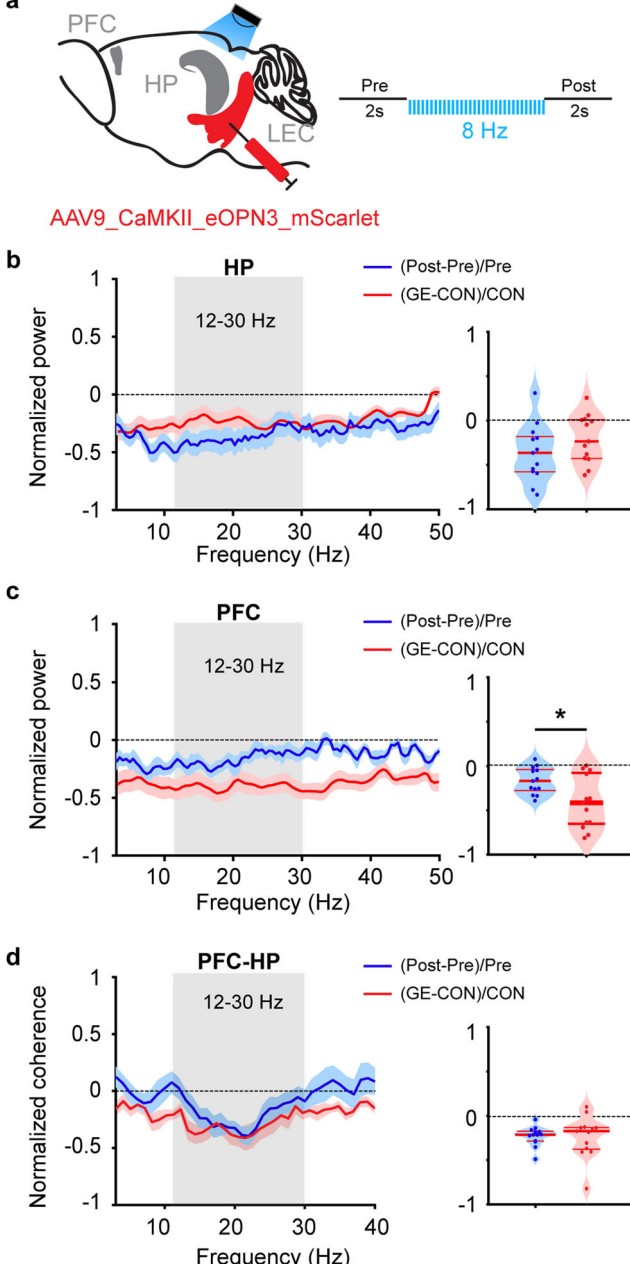

**Fig. 9 Light-induced inhibition of entorhinal terminals in HP of CON mice. a** Schematic of the stimulation protocol of LEC from AAV9_CaMKII_eOPN3_mScarlet-transfected P1 control mice. **b** Left, power of oscillatory activity in HP after stimulation of LEC axons in HP (post) normalized to the activity before the stimulation (pre) in CON (blue). Red line displays the relative HP power in nonstimulated GE mice. Right, violin plot displaying the average power reduction in 12–30 Hz range (one-way ANOVA, $p = 0.158$, $F_{(1, 24)} = 2.12$). **c** Same as **b** for PFC (one-way ANOVA, p = 0.016, $F_{(1, 24)} = 6.75$). **d** Left, line plots depicting the HP-PFC coherence after pulse stimulation of entorhinal terminals in HP (post) normalized to coherence values before stimulation (pre) (blue). Red line displays the relative HP-PFC coherence in GE mice. Right, violin plot displaying the coherence averaged for 12–30 Hz range (Wilcoxon rank-sum test, $p = 0.79$, zval $= -0.27$, ranksum = 176). For **b**, **c**, and **d** the normalized power/coherence measures the power changes related to the activity before the stimulation for light-induced inhibition of entorhinal terminals in HP of CON mice, and to the activity of nonstimulated CON mice for nonstimulated GE mice. For violin plots, blue and red dots correspond to individual animals and the red horizontal lines display the median as well as 25th and 75th percentiles. *$p < 0.05$.

While at neonatal age the direct LEC-PFC communication was largely intact, the LEC-HP communication was weaker as result of sparser and reduced efficiency of axonal projections to excite the hippocampal neurons. The effects of these deficits were detectable also in the downstream area, the PFC. Even if the passive and active membrane properties of entorhinal HP-projecting neurons were largely unaffected in GE mice, the function of axonal projections to HP was impaired. The origin of this impairment is still unknown. One possible source might represent the dysfunction of local circuits in LEC. The cytoarchitecture of neonatal LEC was normal, yet subtle migration and differentiation deficits cannot be excluded as possible mechanisms, especially when considering that *Disc1* gene represents an intracellular hub of developmental processes. Neurons in deep layers of LEC have abnormal passive and active properties that might perturb the intracortical entorhinal connectivity. Another source of entorhinal dysfunction might be weak upstream input due to olfactory deficits. Robust olfactory deficits have been identified in schizophrenia patients and at-risk youth[62]. The shrinkage of the OB and the resulting abnormal olfactory processing have been considered as a byproduct of an early developmental disturbance[63]. Thus, it can be hypothesized that the LEC disturbance and downstream limbic circuitry, at least in part, the result of an early miswiring of olfactory system. Currently, investigations of olfactory processing of mouse models of disease are lacking. Their achievement might open new perspectives for mechanistic understanding of schizophrenia and in the end, for early diagnostic (i.e. biomarkers) and design of therapeutic strategies.

do not actively whisker at neonatal age have already adult-like olfactory abilities[60]. The OB not only processes and forwards the odor information to LEC, but also spontaneously generates early oscillatory activity that activates LEC[40,61]. Therefore, at neonatal age, the entorhinal direct drive to HP as well as direct and indirect to PFC, is controlled by the olfactory system.

In GE mice, LEC-dependent associative memory is impaired already at pre-juvenile age. The network impairment at this age is rather mild. The mechanisms underlying the age-dependent differences and compensatory effects are still unknown. However, we cannot rule out that the use of urethane anesthesia in pre-juvenile mice might contribute to the observed milder impairment. In contrast, a prominent dysfunction within neonatal entorhinal-hippocampal-prefrontal networks has been identified and might be compensated to a certain amount throughout the development (see normal oscillatory power). However, the abnormal firing rates and communication within pre-juvenile circuits attest to circuit miswiring taking place before this age.

## Methods

**Animal Models**. All experiments were performed in compliance with the German laws and the guidelines of the European Community for the use of animals in research and were approved by the local ethical committee (015/17, 015/18). Timed-pregnant C57BL/6 J mice from the animal facility of the University Medical Center Hamburg-Eppendorf were used. The day of vaginal plug detection was defined as gestational day (G) 0.5, whereas the day of birth was defined as postnatal day (P) 0. Multisite extracellular recordings and behavioral testing were performed on pups of both sexes during neonatal development (i.e. P8-P10) as well as during pre-juvenile development (i.e. P16-P23). Heterozygous genetically engineered mutant DISC1 mice carrying a *Disc1* allele (Disc1Tm1Kara) on a C57BL6/J background were used. Due to two termination codons and a premature poly-adenylation site, the allele produces a truncated transcript[64]. Genotypes were determined using genomic DNA and following primer sequences: forward primer 5′-TAGCCACTCTCATTGTCAGC-3′, reverse primer 5′-CCTCATCCCTTCCA

CTCAGC-3′. Mutant DISC1 mice were challenged by MIA, using the viral mimetic poly I:C (5 mg kg$^{-1}$) injected intravenously (i.v.) into the pregnant dams at gestational day G9.5. The resulting offspring mimicking the dual genetic-environmental etiology of mental disorders were classified as GE mice (DISC1 knockdown + MIA). The offspring of wild-type C57BL/6 J dams injected at G9.5 with saline (0.9%, i.v.) were classified as CON mice (control).

**Electrophysiological recordings in vivo.** All neonatal mice were recorded in the absence of anesthesia. Since neonatal mice spent a substantial time sleeping and the sleeping rhythms at this age lack the adult characteristics, we did not distinguish between the states and generally defined the state as "non-anesthetized". For neonatal recordings in non-anesthetized state, 0.5% bupivacain/1% lidocaine was locally applied on the neck muscles. For pre-juvenile recordings under anesthesia, mice were injected intraperitoneally (i.p.) with urethane (1 mg/g body weight; Sigma–Aldrich, MO, USA) prior to surgery. For both age groups, under isoflurane anesthesia (induction: 5%, maintenance: 2.5%) the head of the pup was fixed into a stereotaxic apparatus using two plastic bars mounted on the nasal and occipital bones with dental cement. The bone above the PFC (0.5 mm anterior to bregma, 0.1–0.5 mm right to the midline), hippocampus (3.5 mm posterior to bregma, 3.5 mm right to the midline), LEC (4 mm posterior to bregma, 6 mm right to the midline) was carefully removed by drilling a hole of <0.5 mm in diameter. After a 10 min recovery period on a heating blanket, mouse was placed into the setup for electrophysiological recording. Throughout the surgery and recording session the mouse was positioned on a heating pad with the temperature kept at 37 °C.

A four-shank optoelectrode (NeuroNexus, MI, USA) containing 4 × 4 recording sites (0.4–0.8 MΩ impedance, 100 μm spacing, 125 μm intershank spacing) was inserted into the PL of PFC. A one-shank optoelectrode (NeuroNexus, MI, USA) containing 1 × 16 recordings sites (0.4–0.8 MΩ impedance, 50 μm spacing) was inserted into CA1 area. A one-shank optoelectrode (NeuroNexus, MI, USA) containing 1 × 16 recordings sites (0.4–0.8 MΩ impedance, 100 μm spacing) was vertically inserted into LEC by placing them parallel to the pup's plane. An optical fiber ending 200 μm above the top recording site aligned with each recording shank. A silver wire was inserted into the cerebellum and served as ground and reference electrode. Extracellular signals were band-pass filtered (0.1–9000 Hz) and digitized (32 kHz) with a multichannel extracellular amplifier (Digital Lynx SX; Neuralynx, Bozeman, MO, USA) and the Cheetah acquisition software (Neuralynx). Spontaneous (i.e. not induced by light stimulation) activity was recorded for 20 min at the beginning of each recording session as baseline activity. The position of recording electrodes in the PL, CA1 area of i/vHP and LEC was confirmed postmortem. Wide-field fluorescence images were acquired to reconstruct the recording electrode position in brain slices of electrophysiologically investigated pups. Only pups with correct electrode position were considered for further analysis. In PL, the most medial shank was inserted to target layer 2/3, whereas the most lateral shank was located into layer 5/6. For the analysis of hippocampal LFP, the recording site located in the pyramidal layer, where SPWs reverse[65] was selected to minimize any non-stationary effects of large amplitude events. For the analysis of LEC LFP, the recording site that 700 μm above the pyramidal layer of CA1 was selected.

**Viral transfection in pyramidal neurons of LEC and light stimulation.** Transfection of pyramidal neurons with a ChR2 derivate (100 nl), a targeting-enhanced mosquito homolog of the vertebrate encephalopsin (eOPN3, 50 nl) or hM4D was achieved by injecting the construct AAV9-CaMKII-ChR2(H134R)-mCherry (Addgene, Watertown, MA, USA), AAV9-CaMKII-eOPN3-mScarlet or AAV9-CaMKII-hM4D(Gi)-EGFP (Addgene, Watertown, MA, USA) at a titer > 1 × 10$^{13}$ vg/mL in LEC on the right hemisphere of P1 pup, respectively. In line with pilot experiments using 50, 100, and 150 nl volume, the used volume of 100 nl led to reliable transfection of a large number of neurons confined to LEC. The pups were placed in a stereotactic apparatus and kept under anesthesia with isoflurane (induction: 5%, maintenance: 2.5%) for the entire procedure. A 10 mm incision of the skin on the head was performed with small scissors. The bone above the LEC was carefully removed using a syringe. The injection was achieved via a 10 μl microsyringe pump controller. The injection speed (0.05 μl/min) was slow with the maintenance of the syringe in place for at least 8 min. To stimulate ChR2-expressing neurons, pulsatile (laser on-off, pulse 3 ms-long, 8 Hz, 3 s-long) or ramp (linearly increasing power, 3 s-long) light stimulations were delivered with an arduino uno (Arduino, Italy) controlled diode laser (473 nm; Omicron, Austria). Laser power was adjusted to trigger neuronal spiking in response to >25% of 3 ms-long light pulses at 8 Hz. To stimulate eOPN3-expressing axons, pulsatile (laser on-off, pulse 5 ms-long, 8 Hz, 4 s-long) light stimulations were performed with an arduino uno (Arduino, Italy) controlled diode laser (473 nm; Omicron, Austria). Resulting light power was in the range of 20–40 mW mm$^{-2}$ at the fiber tip (i.e. 3–5 mW mm$^{-2}$ at the probe tip). To chemogenetically inhibit LEC by activating hM4D, DREADD agonist 21 (C21) was i.p. injected (3 mg kg$^{-1}$ body weight).

**In vitro whole-cell patch-clamp recordings.** Whole-cell patch-clamp recordings were performed from neurons identified by their location in the LEC and their projections to PFC (CTB488 / Fluorogold retrograde-labeled neurons) or HP (CTB555 retrograde-labeled neurons). All recordings were performed at room temperature. Recording electrodes (5–8MΩ) were filled with K-gluconate-based solution containing (in mM):130 K-gluconate, 10 Hepes, 0.5 EGTA, 4Mg-ATP, 0.3Na-GTP, 8 NaCl (285 mosmol kg-1H2O, pH 7.4) and 0.5% biocytin for post hoc morphological identification of recorded cells. Capacitance artefacts were minimized using the built-in circuitry of the patch-clamp amplifier (HEKA EPC 10, HEKA Elektronik, Germany). The signals were low-pass filtered at 10 kHz and recorded online. All potentials were corrected for the liquid junction potential of the gluconate-based electrode solution, which, according to own measurement, was −8.65 mV. The resting membrane potential (RMP) was measured immediately after obtaining the whole-cell configuration. For the determination of input resistance (Rin), membrane time constant (Tm) and membrane capacitance (Cm), hyper-polarizing current pulses (−60 pA) of 600 ms in duration were applied from the resting membrane potential. Firing frequency was assessed at a depolarizing current pulse of 100 pA at the same length of 600 ms. Analysis was performed offline using custom-written scripts in the MATLAB environment.

For patch-clamp recordings accompanied by optogenetic stimulation, whole-cell recordings were performed from neurons located in the CA1 area of the HP from P9 or P10 mice that underwent transfection of LEC with AAV9-CaMKII-ChR2(H134R)-mCherry at P1. Two coronal slices including the i/vHP were used per animal. All recordings were performed from pyramidal neurons that were identified according to their shape, spiking pattern, and action potential width. For optogenetic stimulation in vitro, 470 nm light pulses were applied with a CoolLED system (pE-2) attached to the upright microscope. Maximal light output at 470 nm was measured at 10 mW mm$^{-2}$ with optical power meter (Thorlabs, NJ, USA). For stimulation of entorhinal afferents targeting CA1 neurons, light was centered on the stratum lacunosum-moleculare (~150–200 μm below the stratum pyramidale) and light pulses (10 ms, 15 s interval) were repetitively applied for up to 20 times. To block AMPA and NMDA receptors, 10 μM 6-cyano-7-nitroquinoxaline-2, 3-dione (NBQX) and 50 μM amino-5-phosphonovaleric acid (AP5) were added to the bath solution.

Light-evoked EPSCs (eEPSCs) were averaged over 20 stimuli. Their peak amplitude, onset (i.e. delay between light stimulus and time point at which the response speed exceeded 10 pA ms$^{-1}$) and rise-time were calculated. The coefficient of variation (CV) for a given measured variable was defined as the ratio between the standard deviation and the average value of 20 individual responses to light stimulation.

**Behavioral protocols.** The exploratory behavior and recognition memory of CON and GE mice were tested at pre-juvenile age using previously established experimental protocols[66]. Briefly, all behavioral tests were conducted in a custom-made circular white arena, the size of which (D: 34 cm, H: 30 cm) maximized exploratory behavior, while minimizing incidental contact with testing objects[67]. The objects used for testing of associative recognition were six differently shaped, textured and colored, easy to clean items that were provided with magnets to fix them to the bottom of the arena. Object sizes (H: 3 cm, diameter: 1.5–3 cm) were smaller than twice the size of the mouse and did not resemble living stimuli (no eye spots, predator shape). The objects were positioned at 10 cm from the borders and 8 cm from the center of the arena. After every trial the objects and arena were cleaned with 0.1% acetic acid to remove all odors. A black and white CCD camera (VIDEOR TECHNICAL E. Hartig GmbH, Roedermark, Germany) was mounted 100 cm above the arena and connected to a PC via PCI interface serving as frame grabber for video tracking software (Video Mot2 software, TSE Systems GmbH, Bad Homburg, Germany).

Pre-juvenile mice (P16) were allowed to freely explore the testing arena for 10 min. Additionally, the floor area of the arena was digitally subdivided in 8 zones (4 center zones and 4 border zones) using the zone monitor mode of the VideoMot 2 analysis software (VideoMot 2, TSE Systems GmbH). The time spent by pups in center and border zones, as well as the running distance and velocity was quantified.

All protocols for assessing novel object preference (NOP) in P17 mice, novel object preference (distinct objects) (NOPd) in P17 mice and object-location preference (OLP) tasks in P18 mice consisted of familiarization and test trials. During the familiarization trial each mouse was placed into the arena containing two different objects and released with the back to the objects. After 10 min of free exploration of objects the mouse was returned to a temporary holding cage. Subsequently, the test trial was performed after a delay of 5 min post-familiarization. In NOPd task, the mice were allowed to investigate one familiar and one novel object with a different shape and texture for 5 min. The nature of this test is similar to the novel object preference test, except that the test trial involves an association between two different objects (an association of object-object). In OLP task, the mice were allowed to investigate one familiar and a copy of the old object that was previously presented for 5 min. This test examines whether animals recognize the location that was once occupied by a particular object (an association of object-location). Object interaction during the first 4 min was analyzed and compared between the groups. All trials were video-tracked and the analysis was performed using the Video Mot2 analysis software. The object recognition module of the software was used and a 3-point tracking method identified the head, the rear end and the center of gravity of the mouse. Digitally, a circular zone of 1.5 cm was created around each object and every entry of the head point into this area was considered as object interaction. Climbing or sitting on the

object, mirrored by the presence of both head and center of gravity points within the circular zone, were not counted as interactions. Mice with low level of exploration (i.e. <20 cm/min) were excluded from further analysis. The discrimination index was defined as (time at novel object−time at old object) / time at both objects for NOP and NOPd, and as (time at displaced object—time at stationary object) / time at both objects for OLP.

To quantify cFos expression in mouse doing NOPd or OLP task, P16 CON mice were randomly divided into four groups (*n* = 4 mice/group). The mice were allowed to freely explore the arena containing two different objects for 10 mins. This familiarization process continued for 3 days with 2 trials per day. On the third day (P18), 5 min after the last familiarization trial, 3 mice from two groups were signed to perform the test trial (5 min) of NOPd task, whereas one mouse to perform the familiarization trial. Similarly, for the other 2 groups, 3 mice were assigned to perform the test trial (5 min) of OLP task and one mouse to perform the familiarization trial. The mice were perfused ~90 min after the last behavioral trial.

**Retrograde tracing**. For retrograde tracing, P7 mice received retrograde tracer CTB555 (Cholera Toxin Subunit B, Alexa Fluor 455 Conjugate) injections into HP (0.7 mm anterior from the lambda, 2.4 mm from midline, 1.6 mm depth), and CTB488 (Cholera Toxin Subunit B, Alexa Fluor-488 Conjugate) injections into PFC (0.7 mm anterior from to bregma, 0.1 mm from midline, 1.9 mm depth). The pups were placed in a stereotactic apparatus and kept under anesthesia with iso-flurane (induction: 5%, maintenance: 2.5%) for the entire procedure. A 10 mm incision of the skin on the head was performed with small scissors. The bone above the HP and PFC was carefully removed using a syringe. A total volume of 0.1 μl of CTB (2.5% in PBS) was delivered via a 10 μl microsyringe pump controller into PFC or HP. The slow injection speed (0.05 μl/min) and the maintenance of the syringe in place for at least 8 min ensured an optimal diffusion of the tracer. The pups were perfused at P10.

**Anterograde tracing**. To locate the innerved neurons in PFC by LEC, anterograde trans-synaptic tracer wheat germ agglutinin (WGA) (Thermo Fisher Scientific, USA) was used. To locate the LEC innerved neurons in HP, the mixed WGA and Fluorogold solution was used. Mice were injected at P8 with WGA unilaterally into LEC. A total volume of 0.1 μl of WGA (2.5% in PBS) was delivered via a 10 μl microsyringe pump controller. The slow injection speed (0.05 μl/min) and the maintenance of the syringe in place for at least 8 min ensured an optimal diffusion of the tracer. 40 h after the injection, the pups were perfused.

**Histology and staining protocols**. Histological procedures were performed as previously described[26,27]. Briefly, P8-10 and P18-23 mice were anesthetized with 10% ketamine (aniMedica)/2% xylazine (WDT) in 0.9% NaCl solution (10 μg/g body weight, i.p.) and transcardially perfused with Histofix (Carl Roth) containing 4% paraformaldehyde. Brains were postfixed in Histofix for 24 h and sectioned coronally at 50 mm (immunohistochemistry) or 100 mm (quantification for CTB labeled neurons). Free-floating slices were permeabilized and blocked with PBS containing 0.8 % Triton X 100 (Sigma–Aldrich, MO, USA), 5% normal bovine serum (Jackson Immuno Research, PA, USA) and 0.05% sodium azide. For cFos staining, slices were incubated with rabbit anti cFos (1:250, MA5-15055, Ther-moFisher), followed by Alexa Fluor-488 goat anti-rabbit IgG secondary antibody (1:500, A11008, ThermoFisher). For BDA staining, slices were incubated with streptavidin (Cy3, 1:500, 016-160-084, Jackson Immuno Research). For WGA staining, slices were incubated with rabbit anti lectin (1:1000, A2052, Sigma–Aldrich), followed by overnight incubation of biotinylated goat anti-rabbit (1:200, BA-1000, Vector Laboratories), and subsequent 2 h streptavidin (Alexa Fluor-488, 1:500, S11223, ThermoFisher). Slices were transferred to glass slides and covered with Fluoromount (Sigma–Aldrich, MO, USA). Wide-field fluorescence images were acquired to reconstruct the recording electrode position.

**Quantification of cFos, CTB488, CTB555, WGA-labeled neurons**. All quanti-fications were carried out blind to the experimental condition. Using a light microscope, photographs of the relevant areas (LEC for cFos, CTB488, CTB555; PFC for WGA) were taken with a consistent light level (Olympus FX-100). 3~4 slices per animal were used (supplementary table 1). Sections were collected in three equally spaced series. To reduce the redundancy of information in neigh-boring slices, only one of the series was mounted or used for subsequent staining and analysis. Images were processed using ImageJ software. The number of cFos positive neurons, CTB488 positive neurons, CTB555 positive neurons and WGA positive neurons were counted manually in the interested regions.

**Quantification of mCherry-labeled axons**. High magnification images were acquired by confocal microscopy (DM IRBE, Leica Microsystems, Zeiss LSN700) from stratum lacunosum-moleculare of CA1 to quantify LEC axonal terminals labeled by BDA. Microscopic stacks were acquired as 2048 × 2048 pixel images (pixel size, 78 nm; Z-step, 500 nm). All images were similarly processed and ana-lyzed using ImageJ software.

**Data analysis**. Data were imported and analyzed offline using custom-written tools in Matlab software version 7.7 (Mathworks). The data were processed as following: (i) band-pass filtered (500–5000 Hz) to detect MUA as negative deflections exceeding five times the standard deviation of the filtered signals and (ii) low-pass filtered (<1500 Hz) using a third order Butterworth filter before downsampling to 1000 Hz to analyze the LFP. All filtering procedures were per-formed in a phase-preserving manner. The position of DiI-stained recording electrodes in PL (most medial shank confined to layer 2/3, most temporal shank confined to layer 5/6), CA1 and LEC was confirmed postmortem by histological evaluation. Additionally, electrophysiological features (i.e. reversal of LFP and high MUA frequency over stratum pyramidale of CA1) were used for confirmation of the exact recording position in HP.

Discontinuous oscillatory events were dected using a previously developed unsupervised algorithm[68] and confirmed by visual inspection. Briefly, deflections of the root-mean-square of band-pass (3–100 Hz) filtered signals exceeding a variance-depending threshold were assigned as network oscillations. The threshold was determined by a Gaussian fit to the values ranging from 0 to the global maximum of the root-mean-square histogram. Only oscillatory events >1 s were considered for further analysis. Time-frequency plots were calculated by transforming the data using the Morlet continuous wavelet.

For power spectral density analysis in neonatal mice, 1 s-long window of network oscillations was concatenated and the power was calculated using Welch's method with non-overlapping windows. For power calculation in pre-juvenile mice, the continuous oscillatory activity of the entire recording has been used. For optical stimulation, we compared the average power during the 1.5 s-long time window preceding the stimulation to the last 1.5 s-long time window of light-evoked activity.

Single unit activity (SUA) was detected and clustered using klusta and manually curated using phy (https://github.com/cortex-lab/phy). Data were imported and analyzed using custom-written tools in the MATLAB. To calculate firing probability for each single unit, we firstly defined 'light-evoked spiking' as spikes occurring in a 20 ms-long time window after the start of the light pulse. In a second step, the firing probability for a single unit was calculated as the number of light-evoked spikes divided by the total number of the pulses.

Single units were sorted using klusta described above. Spikes from all clustered units were summed up for each mouse. The firing rate was calculated for each animal by dividing the total number of spikes by the duration of the analyzed time window.

Co-occurring oscillatory activities were detected and extracted before coherence was calculated. Coherence was calculated using the coherence method. Briefly, the coherence was calculated (using the functions cpsd.m and pwelch.m) by cross-spectral density between the two signals and normalized by the power spectral density of each. The computation of the coherence C over frequency (f) for the power spectral density P of signal X and Y was performed according to the formula: (1)

$$C_{XY}(f) = \left| \left( \frac{P_{XY}(f)}{\sqrt{P_{XX}(f)P_{YY}(f)}} \right) \right| \quad (1)$$

To investigate the directionality of functional connectivity between PFC and HP, gPDC was used. gPDC is based on linear Granger causality measure in the frequency domain. The method attempts to describe the causal relationship between multivariate time series based on the decomposition of multivariate partial coherence computed from multivariate autoregressive models. The LFP signal was divided into 1s-long segments containing the oscillatory activity. After de-noising using Matlab wavelet toolbox, gPDC was calculated using a previously described algorithm[69,70].

The spatial pattern of light propagation in vivo was estimated using a previously developed model[44] based on Monte Carlo simulation (probe parameters: light fiber diameter: 50 μm, numerical aperture: 0.22, light parameters: 594 nm, 0.6 mW).

**Statistics**. Statistical analyses were performed in Matlab environment. Data were tested for normal distribution using the Levene's test (F-test). Paired *t*-test or one-way ANOVA was performed to detect significant differences when the variance was normally distributed. Otherwise, non-parametric Wilcoxon rank-sum test was used. Investigators were blinded to the group allocation when the quantifications of cFos expression, CTB488 and CTB555 positive neurons in LEC, and WGA positive neurons in PFC were performed. Chi-square test was used to detect the significance difference between two proportions. Data are presented as mean ± sem. Sig-nificance levels of *p* < 0.05 (*), *p* < 0.01 (**), or *p* < 0.001 (***) were tested. Sta-tistical parameters can be found in the main text.

**Reporting summary**. Further information on research design is available in the Nature Research Reporting Summary linked to this article.

## Data availability
LFP and SUA data are available at the following open-access repository: https://gin.g-node.org/xiaxiaxu/Developmental_ephys_LEC-HP-PFC. Further data supporting the findings of this study are available from the corresponding authors on request. Source data are provided with this paper.

## Code availability

All the codes used in the current study are available on github: https://github.com/XiaxiaXu/Toolbox_LFP_Spike.

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

## Acknowledgements

We thank Dr. Joseph Gogos for providing the DISC1 mice and Dr. Simon Wiegert for providing AAV9-CaMKII-eOPN3-mScarlet. We also thank A. Marquardt, C. Tietze, A. Dahlmann, and P. Putthoff for excellent technical assistance. This work was funded by grants from the European Research Council (ERC-2015-CoG 681577 to I.L.H.-O.), the German Research Foundation (SFB 936 B5 and Ha4466/11-1 to I.L.H.-O.), Marie Curie Training Network euSNN (MSCA-ITN-H2020-860563 to I.L.H.-O.), Horizon 2020 DEEPER 101016787 (to I.L.H.-O.), and Landesforschungsförderung Hamburg (LFF76, LFF73 to I.L.H.-O.).

## Author contributions

I.L.H.-O. and X.X. designed the experiments, X.X., L.S., and R.K. carried out the experiments, X.X. analyzed the data, I.L.H.-O. and X.X. interpreted the data and wrote the paper. All authors discussed and commented on the manuscript.

## Funding

## Competing interests

The authors declare no competing interests.
