## [Peer Review File · Nature Communications]

Developmental decrease of entorhinal-hippocampal communication in immune-challenged DISC1 knockdown miceREVIEWER COMMENTS

Reviewer #1 (Remarks to the Author):

This manuscript from Xu and colleagues investigates the functional and anatomical connectivity between the lateral entorhinal cortex (LEC), hippocampus (HP) and prefrontal cortex (PFC) in a mouse model of schizophrenia (GE) and at immature stages.

Their results suggest that there is a decreased functional connectivity between LEC and HP, and between HP and PFC, but not between LEC and PFC in immature GE mice. Interestingly anterograde labelling tracing data show that there are fewer hippocampal projecting neurons in the LEC in this model as compared to controls. Authors conduct a series of optogenetic manipulations that suggest that LEC is less likely to drive HP responses in the GE model and that LEC stimulations, which normally increase HP-PFC coherence in their condition, fails to do so in GE mice. In contrast LEC-PFC connectivity is not affected in these animals.

The authors propose the following scenario: "already at neonatal age, the entorhinal function gating prefrontal-hippocampal circuits is impaired in a mouse model of disease "

In general the approach is interesting and there is a range of exciting techniques and potentially interesting findings. A particularly powerful measure is the generalized partial directed coherence, which reflects a frequency-domain representation of the concept of Granger causality, i.e. a measure of the directionality of network interactions at different frequencies. Performing such experiments in mouse pups is extremely difficult, the questions asked are extremely relevant and the authors have to be congratulated for tackling such challenges.

That said, there are major concerns that need to be addressed in order to assess the validity and meaning of the results

-First of all, the title is not supported by the current data. Authors did not show a direct cause-effect relationship between the so called "entorhinal gate" and HIP-PFC communication. They did not perform any manipulation of the LEC that restores connectivity and behavior in GEN mice nor alterations in CON mice that mimic the GEN findings.

Same for the discussion: p9 lines 341-342: "... (iii) the disruption of entorhinal-hippocampal communication is sufficient to decrease the activation of PFC," this has not been demonstrated experimentally, authors did not decrease such communication in controls.

-A major issue concerns the optogenetic data. First, the amount of viral vectors (0.1ul) injected in pups is extremely important for this age and it looks like, from supp. figure 6, mcherry expression is found beyond the injection site.

Furthermore, the resolution of this figure is very poor and we cannot see any fiber or cell body in the injection sites or the structures of interest. To understand the effects of the optogenetic stimulations it is critical to verify 1) that no cell from the hippocampus was infected 2) that the projection fibers indeed reached HIP and PFC and in a comparable way between groups. The authors show projections via CTB or VGA expression, it is imperative that they also show and quantify mcherry expression. This will ensure that the results are indeed due to differences in innervation rather than differences in the effectiveness of injections and propagation in target structures. For instance it is possible that for unexpected reasons CON animals had more efficient opsyn expression than GE mice or that hippocampal cells were directly stimulated during LEC fiber stimulation.

Please provide current source density analysis of the hippocampal response to LEC stimulation. The sink should be, as expected in the str. Lac. Mol. This can be easily performed with existing data.

Given the concerns about CHR2 expression, can you provide slice experiments showing that the optogenetic response is monosynaptic and blocked by glutamatergic antagonists?

Still concerning optogenetic results. Stimulations are performed at 8Hz and figures indeed show a response from units at the same frequency in LEC and HIP. However, coherence at 8Hz is not affected. Rather the coherence in 10-20Hz is affected. Can the authors explain this shift?

Finally stimulations are performed at 20-40mW which is extremely high. Can the authors show evidence that they "did not cause local tissue heating that might interfere with neuronal spiking". Authors should provide control data, showing LFP data in mice without opsyn.

-What is the rationale for focusing specifically on LEC rather than, for instance on the MEC C-fos experiments show increases of neuronal LEC activation but authors do not show that this increase is specific of the LEC and not observed in other structures. In addition, authors refer to the behavioral task as an "LEC-dependent associative recognition memory" (p5. line 105). Yet, they do not provide data that supports this statement. Showing LEC dependency would require to perform inactivation or lesion experiments, which are not done here.

-Generally, statistical tests in this manuscript deserve a better presentation. First, statistics like t-tests and ANOVAs require homogeneity of variances, from the violin graphs it looks like this is not always the case. Can the authors verify the homogeneity of variance and, if the rule is violated, apply other types of tests, such as non-parametric ones?

On p5 line 115: it is not clear here to which comparison this ANOVA is reporting. Ideally the construction of the anova should test, in the same time 1) the effect of task session, 2) effect of the animal model and 3) interaction. Could you test the interaction significance? Why are the ANOVA degrees of freedom (1,38) when you have 20+16 animals? here it should be rather (1,34). Same for the ANOVA used to compare the power of oscillatory events in pups: the number of CON and GE are, respectively 12 and 10 animals. However, the degrees of freedom of the anova are 28 (it should be 20). p8: line 187-193: please give the F values for the anova. On figure 6b, on the amplitude plot, it clearly appears that outliers (two in each group) drive the statistical significance.

Authors should provide more raw data displaying different aspects of the results that are only shown in average data. For instance it is important to get a better idea of what spontaneous network activity looks like in the GE pup. Since the power of spontaneous oscillations was extremely low in GE mice, it is not clear how they were detected. Similarly coherence data should be supported by LFP traces showing oscillations in both structures

-Firing rate data are not sufficiently explained. For instance in p6 line 148, are we talking about single units, multiunits, pyramidal cells, interneurons or all cells. What is the the number of cells recorded, the number of cells per animals and the number of animals.

minor points:

-Figure 2: why restrict to 4-30Hz while there is clearly a peak in beta-gamma for LEC-HP coherence (20-40hz fog 2d)?

-In several instances, (e.g. suppl data fig 1) quantifications are said to be performed on "3~4 slices". Is it 3 or 4? Could the authors provide exact number, maybe in a table?

To help the reader, authors could better explain what is the single hit E and G immediately rather than in the methods section.

p9 line 229: "... intensity of labelled terminals" the term terminal is misused here. The image resolution used here does not allow to identify the intensity of labelled terminals but of axons.

Alltogether, I believe that these concerns could be addressed rather easily and that the manuscript would be greatly improved by it.

Reviewer #2 (Remarks to the Author):

In this study, the authors examine how the combination of a genetic and an environmental risk factor for schizophrenia (SCZ) disrupts neural activity and coordination within a circuit consisting of the lateral entorhinal cortex (LEC), hippocampus (HPC) and prefrontal cortex (PFC). The authors perform simultaneous recordings from all three structures, as well as anatomical and optogenetic investigations of connections within this circuit, in neonatal mice (postnatal day 8-10) carrying a genetic risk factor for schizophrenia (DISC1 mutation) and which are also exposed to a well-established environmental risk factor for the disease (maternal immune activation), thus modeling the influence of gene-environment (GE) interactions in the pathogenesis of SCZ. The authors find that LEC-HPC and HPC-PFC synchrony is disrupted in GE mice and that they also show deficits in object recognition later in life. The authors are also able to demonstrate that projections between LEC and HPC are weaker in these mice, measured both anatomically and optogenetically, thus providing a possible mechanism for the LEC-HPC synchrony deficits. In contrast, LEC-PFC projections are not disrupted.

In general, the manuscript is clearly written and the results are for the most part convincing. A particular strength of the study is the combination of electrophysiological, anatomical and optogenetic methods, which nicely complement each other and strengthen the authors' conclusions regarding the connectivity deficits in the GE mice. The GE model of SCZ is an additional strength of the study and sets it apart from many others which typically examine genetic and environmental risk factors separately. To my knowledge, this is also the first characterization of LEC-HPC connectivity deficits in a SCZ model and is thus an important and novel contribution to the field, given that HPC dysfunction is well established in the disease but relatively little is known about the neural circuit mechanisms underlying this dysfunction. The study will therefore be of interest to a wide readership. The HPC-PFC synchrony deficits are also interesting, although less novel given that such deficits have already been described by the authors in the GE model (Xu et al., 2021). The authors do suggest that the LEC-HPC deficits contribute to the HPC-PFC deficits - this is for example suggested in the title of the paper - and more generally might contribute to PFC dysfunction - but this is not convincingly supported by their results and would thus require additional experimental data. However, in my opinion the LEC-HPC results are sufficiently interesting and novel on their own so that focusing on these results and exercising more caution when discussing how they relate to the HPC-PFC deficits would be sufficient to make the manuscript suitable for publication. More detailed comments and suggestions for improvement are provided below.

Major comments

The authors have convincingly demonstrated that LEC-HPC communication is disrupted in the GE mice. However, the authors also make the claim that this deficit underlies the deficits in HPC-PFC synchrony observed in these animals. In fact, the title of the paper claims that a "Developmental decrease of entorhinal gate disrupts prefrontal-hippocampal communication" and this is also implied in the abstract: "the entorhinal function gating prefrontal-hippocampal circuits is impaired". However, these claims are not convincingly supported by their results. First of all, it is not clear how the LEC-HPC disruption could lead to a HPC-PFC disruption. The authors claim that the LEC is a "gatekeeper of prefrontal-hippocampal circuits" but it is unclear what the authors mean by this. One possibility could be that the LEC, via its projections to the PFC, might modulate how PFC neurons are influenced by HPC inputs. But in the GE model LEC-PFC inputs appear to be intact so this seems unlikely. The authors also suggest that the LEC-HPC connectivity deficits might impair the ability of LEC to indirectly activate the PFC: "sparser and less efficient projections from LEC cause weaker activation of HP and indirectly, of its target, the PFC". This is indeed supported by the authors' optogenetic stimulation experiments (Figure 7). However, if this were indeed the case one would expect impaired LEC-PFC synchrony in the GE mice, which is not observed (Figure 2). Thus, how the connectivity deficits impact PFC function and HPC-PFC coupling is somewhat inconclusive and the authors should be more cautious in their statements on this issue. However, as I already mentioned, I think the LEC-HPC findings are themselves of sufficient interest and novelty to be the main focus of the manuscript.

The authors perform recordings in GE and control mice at two developmental periods: pre-juvenile

(postnatal day 20-23, similar to the age in which the behavioral tests were performed) and neonate (postnatal day 8-10). In the pre-juvenile animals (sup. Fig. 3), they report no differences in LFP power in the three structures but firing rates are higher in the GE mice. In the neonatal GE mice (Fig. 2), LFP power is reduced in all three structures and LEC-HP as well as HP-PFC synchrony is reduced. Firing rates are not reported. These are two nice datasets but they are not presented in an ideal way and their divergent results are not discussed sufficiently. First, the pre-juvenile dataset is most relevant to the behavioral deficits since this is the age range at which the behavioral deficits were uncovered. I therefore think this supplementary figure deserves to be a main figure. Second, the authors should analyze the two datasets in exactly the same way so that they can be compared: synchrony should be quantified in the pre-juveniles and firing rates should be quantified in the neonates. Third, it is not clear why LFP power is reduced in neonatal GE mice but not in the older pre-juvenile GE mice. This is not discussed at all but of course requires an explanation: why would a deficit in early development disappear at a later developmental stage? This should be discussed after the two datasets have been analyzed in the same way in case additional differences between the results of the two datasets emerge.

Minor comments

Title: "Developmental decrease of entorhinal gate disrupts prefrontal-hippocampal communication" - as discussed above, a causal relationship between the entorhinal and the HPC-PFC deficits is not supported by the authors' data. The title should be rewritten to reflect the main findings of the study, i.e. the LEC-HPC connectivity deficits.

Line 51: "... the tasks ..." - not clear what kind of tasks is being referred to here

The NOPd test is presented as a test of "associative" memory but I don't see what association is being tested. In the testing phase, one of the two objects from the familiarization phase is replaced with a new one. The fact that the mice explore this object more simply shows that they recognize it as novel, not that they have formed any object-object association. It is of course possible that the animals form an association between the two objects in the familiarization phase and this shapes their exploratory behavior during the testing phase, but the current behavioral paradigm cannot distinguish this from the simpler non-associative novel object recognition). The OLP task, on the other hand, is associative. However, the deficit in the non-associative NOPd task in the GE mice suggests that the OLP deficit in these mice may reflect a more general deficit in object recognition.

Line 142: "multiple-unit activity (MUA)" - is this multiple SUAs or "multi-unit" activity (i.e. unsorted spikes)?

Line 149: I see no reason to report logarithmic values of firing rates, this is unusual and makes the results less transparent. Also the base of the logarithm is not reported so it is not possible to convert these values to Hz. Please report all firing rates in spikes per second.

General: Indicate what error bars in figures represent and over what (animals? cells?) they are calculated.

Based on Figure 4, claims are made about the effectiveness of optogenetic LEC activation in driving LEC neurons (GE=CON) and HPC neurons (GE < CON). However, the authors only show single cell examples and no statistical tests are given to back up their statements. This would be an important result so the authors need to show the distribution of responses in both genotypes and perform the appropriate statistical tests.

Figure 5c: what does "Firing probability" mean? Also the units are very different than in the similar Figure 6c. I also had the same question about the numbers mentioned when quantifying the efficiency of light stimulation (line 293).

Line 310-311: The different ratios of excited cells in CON and GE mice - is this significant? Does the fraction of inhibited cells differ between the two groups?

Figure 5a: show at higher magnification so that the shape of individual neurons can be visualized

Figure 6 and Figure 5 present the results of the same experiment done in different parts of the circuit (LEC-PFC vs LEC-HPC), but the figures differ in their presentation. Please make them identical to facilitate comparison between these two experiments.

In Figures 5 and 6, the responses are described as having an onset of 15 ms but I think the authors are referring to the latency of the maximum negativity of the response. It seems clear from the example traces that the onset of the response is much more rapid, as would be expected since it should only reflect the synaptic delay following terminal stimulation.

Figure 5c and 6c: are these individual neuron examples or averages over multiple neurons?

In Figure 5d and 6d and 6e raster plots of neuronal responses to light stimulation are shown. It seems that each line of the raster plot is the response of one neuron. However, raster plots are usually shown for one neuron over multiple trials. It is difficult to make sense of a raster plot where each row is a different neuron (presumably also measured on different trials). I would therefore suggest the following to improve the presentation of the results. First, show a raster plot of at least one example neuron (as, for example, in the author's previous study Chini et al. 2020, Fig. 2). Second, show the average response of each neuron to the light stimulus, sorted by their response magnitude and direction (this seems to have been done already for the raster plots), as a color plot.

Figure 6d and e, right. It is difficult to see some of the data points. Make them larger. It is also not clear why the points have different sizes; if this is for indicating significance it is not necessary since this is also indicated by the y-axis.

Figure 7b: Are the graphs on the left examples of individual neurons? Please make clear. In these graphs, the maximum firing rate reaches ~ 1.5 whereas the graphs on the right report much stronger effects, 1.9 and 1.7.

Line 321: "relative change" - does this mean the percentage change?

Line 332: should be "hypothesized to occur during development" or something similar.

Line 424: the neonatal animals are said to have been recorded in a "non-anesthetized state" - what does this mean exactly? Are the animals awake? Then the state should be described as such.

Line 442: how were electrodes inserted horizontally into the LEC? This would seem to require removing a considerable amount of tissue on the side of the head. It is difficult to see how this can be achieved without anesthesia

Line 355: "giving" should be "given"

Line 363: "Under normal conditions, the neonatal LEC acts as gatekeeper of HP and PFC". It is not clear what the authors mean by 'gatekeeper', they should define precisely what they mean in terms of physiology.

Line 387-8: "Even if the passive and active membrane properties of entorhinal HP-projecting neurons were largely unaffected in GE mice, the neuronal function was impaired." - please explain what is meant by "neuronal function". As far as I can understand they have reported that oscillations in LEC are impaired (Figure 2A) and that they form weaker connections with the HP.

Reviewer #3 (Remarks to the Author):

The article by Xu et al. details a series of experiments that examine LEC-HP-PFC dysfunction as a result of combined genetic and environmental factors in early development (GE). This GE model of

psychiatric disorders is examined using sophisticated genetic tools that allow the authors to provide a very detailed account of how the LEC-HP-PFC network is affected in this model in terms of anatomy, electrophysiology and behavior. In general, the article provides compelling evidence that LEC- CA1 projections are compromised in the model where as LEC-PFC projections are much less affected. In general I think this is an interesting article but there are a number of issues that the authors need to address.

1. The rationale of the studies needs to be more precise. The GE model is cited at the end of the abstract as a 'model of disease.' This needs to be much more specific. The introduction continues in this vein by saying that the major burden of major psychiatric disorders is lifelong cognitive disability. This is also very non-specific. The introduction then becomes more focused on schizophrenia which makes sense so I would recommend that the article is reframed specifically in terms of schizophrenia (rather than 'disease') and the associated cognitive deficits and underlying neural mechanisms.

2. Pseudo-replication. There are a number of places in the manuscript where data are reported at the level of the slice rather than the animal. Worryingly, the figures suggest that analyses have also been carried out using slices as independent subjects. This will falsely increase statistical power. This is the case in the Fos study (Supp Fig 1) and also figures 3b/c and supp 5. These figures and analyses need to be replaced with analyses at the level of the animal.

3. This issue is compounded by the fact that no information is given about sampling in these anatomical studies. Is every section counted? If so, how do the authors guard against counting the same cells in multiple sections? There is also no information given about how sections within regions are chosen. Given that different regions of LEC/HPC/PFC have different anatomical connectivity it is vital that the authors detail how they ensure they are consistently assessing the same regions within each area across mice.

4. There are some inconsistencies between numbers of animals reported, those presented in figures and those used for analysis. As an example, lines 106 says that 20 CON and 16 GE mice were used for the behavioral studies. However, Figure 1a has between 22-24 individual dots (hard to see as they sometimes overlap but a minimum of 22) for the control animals. The relevant ANOVA has degrees of freedom of (1,38) which shows that 40 animals were included in the analysis. None of these match up. The authors need to carefully check numbers cited for methods, figures and analyses throughout the manuscript.

5. The first result reported is that young GE mice have poorer associative recognition memory. While the results of the OLP are consistent with this the NOPd are not as this is not a test of associative recognition. Mice can solve this based entirely on the fact that a novel object is presented. A simple object novelty signal (without the need to remember the association of 2 objects) could support the behaviour of the control mice.

6. The discrimination index is defined as $(\text{time at object 2} - \text{time at object 1}) / (\text{time at object 1} + \text{time at object 2})$. This suggests that the identity of the novel object was not counterbalanced and object 2 was always the novel object. If this is the case how do the authors control for different levels of motivation to explore different objects? All details of counterbalancing need to be supplied.

7. Total levels of exploration in both familiarization and test trials need to be presented and analysed to ensure that the data are not affected by differential encoding or differential motivation to explore novelty.

8. Was there a minimum level of exploration required in familiarization and test phases? Low exploration in familiarization will lead to poor encoding and discrimination indices in the test trial can be skewed by low levels of exploration. It is standard to have a minimum level of exploration for each trial.

9. How accurate is the tracking? How many frames are missed and is there a level at which trials are disregarded due to poor tracking?

10. How do the findings in Figure 3 relate to the Layer 2a and 2b distinction described by Leitner et al. 2016 (Nat Neurosci) and Vandrey et al. 2020 (Current Biology).

Replay to reviewer's comments

Reviewer #1:

In general the approach is interesting and there is a range of exciting techniques and potentially interesting findings. A particularly powerful measure is the generalized partial directed coherence, which reflects a frequency-domain representation of the concept of Granger causality, i.e. a measure of the directionality of network interactions at different frequencies. Performing such experiments in mouse pups is extremely difficult, the questions asked are extremely relevant and the authors have to be congratulated for tackling such challenges.

We thank the reviewer for the constructive feedback and most helpful comments and suggestions.

That said, there are major concerns that need to be addressed in order to assess the validity and meaning of the results.

First of all, the title is not supported by the current data. Authors did not show a direct cause-effect relationship between the so called "entorhinal gate" and HIP-PFC communication. They did not perform any manipulation of the LEC that restores connectivity and behavior in GEN mice nor alterations in CON mice that mimic the GEN findings.

Same for the discussion: p9 lines 341-342: "... (iii) the disruption of entorhinal-hippocampal communication is sufficient to decrease the activation of PFC," this has not been demonstrated experimentally, authors did not decrease such communication in controls.

In line with the reviewer's suggestion, we performed additional experiments to strengthen the direct causal evidence of LEC impact on hippocampal-prefrontal communication. For this, we capitalized on a recently available optogenetic tool, the targeting-enhanced mosquito homolog of the vertebrate encephalopsin (eOPN3), that selectively suppresses neurotransmitter release at presynaptic terminals with high spatiotemporal precision (Mahn et al. 2021). Selective silencing of LEC axons in HP of CON mice mimics the reduced entorhinal drive to HP that we found in GE mice (see figure below). The 12-30 Hz power in HP as well as the HP-PFC coupling, and subsequently the PFC power were reduced after activation of eOPN3 by light, reaching similar values to those reported for non-manipulated GE mice.

We added the new data to the manuscript (Results, page 13, line 355-371) and displayed them in a new figure (Figure 9). Moreover, since no adequate tools to achieve the recovery of developing entorhinal-hippocampal connectivity and function are available, we modified the title to better mirror the experimental data.

(a) Schematic of the stimulation protocol of LEC from AAV9_CaMKII_eOPN3_mScarlet-transfected P1 control mice. (b) Left, power of oscillatory activity in HP after stimulation of LEC axons (post) in HP normalized to the activity before the stimulation (pre) in CON (blue). Red line displays the relative HP power in non-stimulated GE mice. Right, violin plot displaying the average power reduction in 12-30 Hz range. (c) Same as (b) for PFC. (d) Left, line plots depicting the HP-PFC coherence after pulse stimulation of entorhinal terminals in HP (post) normalized to coherence values before stimulation (pre) (blue). Red line displays the relative HP-PFC coherence in GE mice. Right, violin plot displaying the average coherence in 12-30 Hz range. For violin plots, each dot corresponds to a mouse and the red horizontal lines display the median as well as 25th and 75th percentiles.

A major issue concerns the optogenetic data. First, the amount of viral vectors (0.1ul) injected in pups is extremely important for this age and it looks like, from supp. figure 6, mcherry expression is found beyond the injection site. Furthermore, the resolution of this figure is very poor and we cannot see any fiber or cell body in the injection sites or the structures of interest. To understand the effects of the optogenetic stimulations it is critical to verify 1) that no cell from the hippocampus was infected 2) that the projection fibers indeed reached HIP and PFC and in a comparable way between groups. The authors show projections via CTB or WGA expression, it is imperative that they also show and quantify mcherry expression. This will ensure that the results are indeed due to differences in innervation rather than differences in the effectiveness of injections and propagation in target structures. For instance it is possible that for unexpected reasons CON animals had more efficient opsyn expression than GE mice or that hippocampal cells were directly stimulated during LEC fiber stimulation.

Please provide current source density analysis of the hippocampal response to LEC stimulation. The sink should be, as expected in the str. Lac. Mol. This can be easily performed with existing data.

To address the important concern of selective transfection, we performed pilot experiments with different amounts of viral vectors (50 nl, 100 nl, 150 nl). While for both 50 and 100 nl, the transfected neurons were confined to LEC, their number augmented with increasing volume. For the larger amount that we tested (150 nl), few hippocampal neurons were also

transfected. Therefore, all experiments were performed with an amount of 100 nl viral vector that is comparable with the volume used for subiculum and CA1 (Pederick DT et al., 2021) and smaller than the volume used for neocortical areas at the same age (Wong, F.K. et al., 2018; Chen, CC et al., 2018; Modol L et al., 2020). On the other hand, the results of CTB injections (Fig. 4) showed that HP- and PFC-projecting neurons are mainly located in the superficial layers of LEC. Therefore, to target these neurons and avoid transfecting deep located hippocampal neurons, the injection depth did not surpass 0.2 mm. By controlling the injection volume and depth, we mainly transfected neurons located in the superficial layers (see figure below, a, b, c). Very few mCherry-positive neurons were detected in deep layers (d, e) and none was present in HP (f).

AAV9-CaMKII-ChR2-mCherry injection in the LEC (middle) led to mCherry-labeled neurons in superficial layer of LEC (a, b, c), deep layer of LEC (d), axons from LEC to HP (e) and no transfected neurons in HP (f).

Furthermore, mCherry-labeled entorhinal axons were found in both HP and PFC (Fig. 6b, Fig. 7b). Quantification of these labeled axons in CON and GE mice showed that fewer axons were present in the PFC when compares to HP in both CON and GE mice. These data are in line with the lower coherence values for LEC-PFC when compared with LEC-HP (Fig. 3c). Moreover, the density of mCherry-labeled axons was lower in the HP of GE mice than CON mice. We illustrated the new data in an additional figure (supplementary fig. 8 and supplementary fig. 11) and describe them in the manuscript (Results, page 10, line 271, 283-284; Methods, page 19, line 513-515)

Following the reviewer's suggestion, we performed current source density analysis of the hippocampal response to LEC stimulation. The sink was detected ~150-200 μm below the *stratum pyramidale*. This layer corresponds to *stratum lacunosum*, where also the LEC

terminals accumulated (Fig. 7b). We added the results of CSD analysis to the manuscript (Results, page 13, line 337-339) and illustrated them in Fig. 7d.

Given the concerns about CHR2 expression, can you provide slice experiments showing that the optogenetic response is monosynaptic and blocked by glutamatergic antagonists?

To address the concern, we conducted patch-clamp recordings from CA1 neurons *in vitro*, during optogenetic activation of LEC axons. We confirmed that the response is monosynaptic and glutamatergic based on following experimental evidence (see figure below): (i) the onset of light-evoked response was short (< 9 ms), (ii) the responses were blocked by bath application of glutamatergic antagonists (AP5+NBQX), and (iii) due to the use of CamKII-alpha promoter, only excitatory pyramidal neurons in LEC have been transfected. We analyzed the properties of light-evoked excitatory postsynaptic current (eEPSC) in GE and CON mice. Less CA1 neurons innervated by LEC have been detected in slices from neonatal GE mice (CON, 16/26; GE, 9/42, $p=0.0021$, $\chi^2 = 9.45$, Chi-square test). The amplitude of light-evoked response was significantly smaller and their rise-time longer in GE mice when compared with CON mice (see figure below). We added the new data to the manuscript (Results, page 12, line 318-332) and illustrated them in an additional figure (supplementary fig.12).

(a) Left, representative image showing ChR2 (H134R) (red) expression in a DAPI-stained coronal slice from a P10 CON mouse following LEC transfection at P1. Right, confocal image showing biocytin-filled CA1 neurons in HP from a P10 CON mouse displayed together with a schematic of light stimulation / recording protocol. (b) Representative light-evoked responses (blue vertical bar, 10 ms) recorded in a CA1 pyramidal neuron (black trace) at -65 mV. The response was abolished by bath application of NBQX and AP5 (purple trace). (c) Average light-evoked responses HP neurons from CON (n=16) and GE (n=9) neurons. Inset, bar diagram of the percentage of responsive CA1 neurons. (d) Violin plots displaying the amplitude (left), coefficient of variation of amplitudes (middle), and rise time (right) of light-evoked responses averaged for all CA1 neurons from CON (n=16) and GE (n=9) neurons. Dots correspond to investigated neurons and the median as well as 25th and 75th percentiles have been considered. * $p < 0.05$, ** $p < 0.01$.

Still concerning optogenetic results. Stimulations are performed at 8Hz and figures indeed show a response from units at the same frequency in LEC and HIP. However, coherence at

8Hz is not affected. Rather the coherence in 10-20Hz is affected. Can the authors explain this shift?

The apparent discrepancy results from the stimulation approach that has been selected to optimally fit the aim of investigation. To test the responsiveness of entorhinal neurons in CON and GE mice, we used pulsed light stimulation. In line with previous results (Bitzenhofer et al., 2017, Ahlbeck et al., 2018) that showed that 8 Hz stimulation reliably activated the cortical neurons in the developing brain, the pulsed stimulation at this frequency led to similar spiking of entorhinal neurons in CON and GE mice (Fig. 5b). Thus, the responsiveness of LEC neurons to light was not impaired in GE mice. However, setting the light stimulation at a given frequency (e.g. 8 Hz) is not suitable when addressing the question, whether the entorhinal drive similarly impacts the hippocampal activity in CON and GE mice. For this aim, we used ramp light stimulations that, as shown in the past (Bitzenhofer et al., 2017; Bitzenhofer et al., 2021), not only leads to less artefacts but also enable neurons to fire at a preferred and not set frequency upon stimulation. The ramp stimulation augmented the LEC-HP coherence within 10-20 Hz in CON but not GE mice, suggesting that that the entorhinal excitatory drive boosts the coupling within this range. We rephrased the text to better highlight the aims of the two experimental strategies.

Finally, stimulations are performed at 20-40mW which is extremely high. Can the authors show evidence that they “did not cause local tissue heating that might interfere with neuronal spiking”. Authors should provide control data, showing LFP data in mice without opsin.

To address the concern, we specified that the light power of 20-40 mW was measured at the fiber tip and corresponds to 3-5 mW at the probe tip. To verify that this light power that is in the range usually used for in vivo manipulations, does not cause local tissue heating and interfere with neural spiking, we performed two investigations. First, we used the previously developed model based on Monte Carlo simulation (Stujenske et al. 2015) to estimate the light-induced temperature increase in the brain tissue. We calculated an increase of ~0.75 °C

during the pulse stimulation (supplementary fig.10 a) that is in the range reported to not cause tissue heating. Second, we performed new experiments and used the same light power in control mice transfected with constructs lacking the opsin (n=3 mice). The firing of LEC, HP, or PFC did not change during stimulation, indicating the lack of heating-induced modulation of firing. We added the new data to the manuscript (Results, page 10, line 275-276) and illustrated them in an additional figure (supplementary fig.10).

(a) Heat map of temperature changes caused by the 8 Hz pulse stimulation estimated by the Monte Carlo model. (b) Representative raster plot of LEC firing activity in response to 30 sweeps of 8 Hz-pulsed stimulation (3 ms pulse length, 473 nm) in LEC of one P9 CON mice. (c, d) Same as (b) for HP and PFC, respectively.

What is the rationale for focusing specifically on LEC rather than, for instance on the MEC. C-fos experiments show increases of neuronal LEC activation but authors do not show that this increase is specific of the LEC and not observed in other structures. In addition, authors refer to the behavioral task as an “LEC-dependent associative recognition memory” (p5. line 105). Yet, they do not provide data that supports this statement. Showing LEC dependency would require to perform inactivation or lesion experiments, which are not done here.

The focus on LEC and not MEC was justified by previous results that showed (i) the tight role of LEC for the entrainment of prefrontal-hippocampal circuits throughout development (Hartung et al., 2016), and (ii) unlike the MEC, the LEC is not critical for spatial navigation but for associating objects and environmental features (Wilson et al., 2013). While excitotoxic lesions have been performed in adult mice and revealed that LEC is critical for associative recognition memory (Wilson et al., 2013), we directly addressed the role of pre-juvenile LEC for this behavioral ability by chemogenetic inactivation. We performed additional experiments and injected AAV9_CaMKII_hM4Di_EGFP in LEC in P1 CON mice. The neuronal activity in LEC was reduced by exposure to the DREADD agonist 21 (compound 21, C21) 45 mins before the NOPd (P17) or OLP (P18) tasks. Similar to the poor performance of GE mice, after C21 injection CON mice were not able to recognize the novelty in both NOPd and OLP test phases. These data show that LEC was not only involved in, but also necessary for these two tasks.

Poor performance of CON mice in NOPd and OLP tasks after silencing of LEC activity by the DREADD agonist 21 (compound 21, C21). (a) Violin plots displaying the discrimination ratio in familiarization and test trials of NOPd task. (b) Same as (a) for OLP task. Black and red dots correspond to individual animals and the red horizontal lines display the median as well as 25th and 75th percentiles.

We added the new data to the manuscript (Results, page 5, line 115-120) and illustrated them in a new supplementary figure (Supplementary fig. 2). Moreover, we discussed in more detail the reason why the study focuses on LEC and not MEC (Discussion, page 15, line 389-397).

Generally, statistical tests in this manuscript deserve a better presentation. First, statistics like t-tests and ANOVAs require homogeneity of variances, from the violin graphs it looks like this is not always the case. Can the authors verify the homogeneity of variance and, if the rule is violated, apply other types of tests, such as non-parametric ones?

In line with the reviewer’s suggestion, we firstly verified the homogeneity of variances through Kolmogorov-Smirnov test. Nonparametric statistical tests (Wilcoxon rank sum test)

were used when the variance is not normally distributed. This information was added to Methods (Methods, page 27, line 712-715)

On p5 line 115: it is not clear here to which comparison this ANOVA is reporting. Ideally the construction of the anova should test, in the same time 1) the effect of task session, 2) effect of the animal model and 3) interaction. Could you test the interaction significance?

We performed three-Way ANOVA to compare the time spent interacting with the novel object and the familiar one for NOPd task. The 3 factors are groups (CON, GE), objects (familiar object, novel object) and tasks (familiarization, test). There are significant differences of the effect of the animal models ($p=0.04$) and objects ($p=0.005$). No significant differences of the effect of task sessions has been detected ($p=0.12$). The interaction between groups and objects ($p=0.031$) as well as between objects and tasks ($p=0.035$) is significant, whereas the interaction between groups and tasks was did not reach significance level ($p=0.90$).

To tested whether mice spend equal time on exploring the two objects during the familiarization session and whether they spend longer time on exploring the novel objects in test session, we applied the paired-sample t-test separately for the familiarization and test session (Fig.1). We added the corresponding values of t and df to the text (page 5, line 125-126, 131-134).

Why are the ANOVA degrees of freedom (1,38) when you have 20+16 animals? here it should be rather (1,34). Same for the ANOVA used to compare the power of oscillatory events in pups: the number of CON and GE are, respectively 12 and 10 animals. However, the degrees of freedom of the anova are 28 (it should be 20). p8: line 187-193: please give the F values for the anova.

We re-performed the statistical analyses and corrected, where necessary, the degrees of freedom and F values.

On figure 6b, on the amplitude plot, it clearly appears that outliers (two in each group) drive the statistical significance.

Using the 1.5IQR rule (i.e. an outlier in a distribution is a number that is more than 1.5 times the length of the box away from either the first (25%) or the third (75%) quartiles), none of the data points was detected as outlier and therefore, cannot be removed. Since the data are not normally distributed, as shown by the Kolmogorov-Smirnov test, we used Wilcoxon rank sum test to detect significant differences (Results, page 13, line 336).

Authors should provide more raw data displaying different aspects of the results that are only shown in average data. For instance it is important to get a better idea of what spontaneous network activity looks like in the GE pup. Since the power of spontaneous oscillations was extremely low in GE mice, it is not clear how they were detected. Similarly, coherence data should be supported by LFP traces showing oscillations in both structures.

Even if raw data from GE mice of comparable age have been previously published by our group (e.g. Hartung et al., 2016; Chini et al., 2020; Xu et al. 2021), we followed the reviewer's suggestion and added representative LFP and MUA traces from all three investigated brain

areas to the revised manuscript (supplementary fig.5). Same rule was used for detecting the oscillatory events in CON and GE mice (Method, page 26, line 674-679). Co-occurring oscillatory events were detected before coherence was calculated. We added this information to the manuscript (Method, page 26, line 695-696).

(a) Extracellular LFP recordings of discontinuous oscillatory activity in the LEC of a P9 CON mouse (left) and a P9 GE mouse (right) displayed after bandpass (2-100 Hz) filtering (top) together with the corresponding MUA (500-5000 Hz) (bottom). Traces are accompanied by the color-coded wavelet spectra of the LFP at identical time scale. (b, c) Same as (a) for HP and PFC, respectively.

Firing rate data are not sufficiently explained. For instance in p6 line 148, are we talking about single units, multiunits, pyramidal cells, interneurons or all cells. What is the number of cells recorded, the number of cells per animals and the number of animals.

To avoid confusions, we replaced the MUA data with single unit activity (SUA) data. For this we re-analyzed the data after spike sorting. Spikes from all clustered units were summed up for each mouse. The firing rate was calculated for each animal by dividing the total number of spikes by the duration of the analyzed time window. As requested, we specified the number of investigated mice.

Minor points:

Figure 2: why restrict to 4-30Hz while there is clearly a peak in beta-gamma for LEC-HP coherence (20-40hz fig 2d)?

As suggested, we tested for statistical significance for the 20-40 Hz range.

In several instances, (e.g. suppl data fig 1) quantifications are said to be performed on “3~4

slices". Is it 3 or 4? Could the authors provide exact number, maybe in a table?

We added the requested table (Suppl. Table1).

To help the reader, authors could better explain what is the single hit E and G immediately rather than in the methods section.

We defined the single-hit E and G mice in the Introduction (page 4, line 89).

p9 line 229: "... intensity of labelled terminals" the term terminal is misused here. The image resolution used here does not allow to identify the intensity of labelled terminals but of axons.

We corrected the term and replaced 'terminals' by 'projections'.

References:

Wong FK, Bercsenyi K, Sreenivasan V, Portalés A, Fernández-Otero M, Marín O (2018). Pyramidal cell regulation of interneuron survival sculpts cortical networks. *Nature* 557, 668–673.

Chen CC, Lu J, Yang R, Ding JB, Zuo Y (2018). Selective activation of parvalbumin interneurons prevents stress-induced synapse loss and perceptual defects. *Mol Psychiatry* 23(7):1614-1625.

Modol L, Bollmann Y, Tressard T, Baude A, Che A, Duan ZRS, Babij R, De Marco García NV, Cossart R (2020). Assemblies of Perisomatic GABAergic Neurons in the Developing Barrel Cortex. *Neuron* 105(1):93-105.e4.

Bitzenhofer SH, Ahlbeck J, Hanganu-Opatz IL. Methodological Approach for Optogenetic Manipulation of Neonatal Neuronal Networks. *Front Cell Neurosci* 11, 239 (2017).

Bitzenhofer SH, Pöplau JA, Chini M, Marquardt A, Hanganu-Opatz IL. A transient developmental increase in prefrontal activity alters network maturation and causes cognitive dysfunction in adult mice. *Neuron* 109, 1350-1364.e1356 (2021).

Ahlbeck J, Song L, Chini M, Bitzenhofer SH, Hanganu-Opatz IL. Glutamatergic drive along the septo-temporal axis of hippocampus boosts prelimbic oscillations in the neonatal mouse. *eLife* 7, e33158 (2018).

Mahn, M., I. Saraf-Sinik, P. Patil, M. Pulin, E. Bitton, N. Karalis, F. Bruentgens, S. Palgi, A. Gat, J. Dine, J. Wietek, I. Davidi, R. Levy, A. Litvin, F. Zhou, K. Sauter, P. Soba, D. Schmitz, A. Lüthi, B. R. Rost, J. S. Wiegert and O. Yizhar (2021). "Efficient optogenetic silencing of neurotransmitter release with a mosquito rhodopsin." *Neuron* 109(10): 1621-1635.e1628.

Stujenske, J. M., T. Spellman and J. A. Gordon (2015). "Modeling the Spatiotemporal Dynamics of Light and Heat Propagation for In Vivo Optogenetics." *Cell Rep* 12(3): 525-534.

Wilson DIG, Watanabe S, Milner H, Ainge JA. Lateral entorhinal cortex is necessary for associative but not nonassociative recognition memory. *Hippocampus* 23, 1280-1290 (2013).

Hartung H, et al. From Shortage to Surge: A Developmental Switch in Hippocampal–Prefrontal Coupling in a Gene–Environment Model of Neuropsychiatric Disorders. *Cereb Cortex* 26, 4265-4281 (2016).

Xu X, Song L, Hanganu-Opatz IL. Knock-Down of Hippocampal DISC1 in Immune-Challenged Mice Impairs the Prefrontal–Hippocampal Coupling and the Cognitive Performance Throughout Development. *Cereb Cortex* 31, 1240-1258 (2021).

Chini M, et al. Resolving and Rescuing Developmental Miswiring in a Mouse Model of Cognitive Impairment. *Neuron* 105, 60-74.e67 (2020).

Pederick DT, Lui JH, Gingrich EC, Xu C, Wagner MJ, Liu Y, He Z, Quake SR, Luo L (2021). Reciprocal repulsions instruct the precise assembly of parallel hippocampal networks. *Science* 372 (6546):1068-1073.

Reviewer #2:

In general, the manuscript is clearly written and the results are for the most part convincing. A particular strength of the study is the combination of electrophysiological, anatomical and optogenetic methods, which nicely complement each other and strengthen the authors' conclusions regarding the connectivity deficits in the GE mice. The GE model of SCZ is an additional strength of the study and sets it apart from many others which typically examine genetic and environmental risk factors separately. To my knowledge, this is also the first characterization of LEC-HPC connectivity deficits in a SCZ model and is thus an important and novel contribution to the field, given that HPC dysfunction is well established in the disease but relatively little is known about the neural circuit mechanisms underlying this dysfunction.

We thank the reviewer for the constructive feedback and most helpful comments and suggestions.

Major comments

The authors have convincingly demonstrated that LEC-HPC communication is disrupted in the GE mice. However, the authors also make the claim that this deficit underlies the deficits in HPC-PFC synchrony observed in these animals. In fact, the title of the paper claims that a "Developmental decrease of entorhinal gate disrupts prefrontal-hippocampal communication" and this is also implied in the abstract: "the entorhinal function gating prefrontal-hippocampal circuits is impaired". However, these claims are not convincingly supported by their results. First of all, it is not clear how the LEC-HPC disruption could lead to a HPC-PFC disruption. The authors claim that the LEC is a "gatekeeper of prefrontal-hippocampal circuits" but it is unclear what the authors mean by this. One possibility could be that the LEC, via its projections to the PFC, might modulate how PFC neurons are influenced by HPC inputs. But in the GE model LEC-PFC inputs appear to be intact so this seems unlikely. The authors also suggest that the LEC-HPC connectivity deficits might impair the ability of LEC to indirectly activate the PFC: "sparser and less efficient projections from LEC cause weaker activation of HP and indirectly, of its target, the PFC". This is indeed supported by the authors' optogenetic stimulation experiments (Figure 7). However, if this were indeed the case one would expect impaired LEC-PFC synchrony in the GE mice, which is not observed (Figure 2). Thus, how the connectivity deficits impact PFC function and HPC-PFC coupling is somewhat inconclusive and the authors should be more cautious in their statements on this issue. However, as I already mentioned, I think the LEC-HPC findings are themselves of sufficient interest and novelty to be the main focus of the manuscript.

We followed the recommendation of the reviewer and focused on the abnormal LEC-HP communication in GE mice. Correspondingly, we modified the title of the manuscript. Moreover, we toned down the statements regarding the role of LEC for the PFC-HP communication and discussed in more detail the pathways of coupling, aiming to explain the coherence results (Discussion, page 15, line 407-417). The coupling between LEC and PFC might occur via 3 distinct pathways: (i) monosynaptic connection from LEC to PFC, (ii) bi-synaptic transmission through direct synaptic connection from LEC to HP neurons, which further directly project to PFC, and (iii) polysynaptic transmission through direct synaptic connection from LEC to HP neurons, which do not directly further project to PFC but through

interplay with other hippocampal neurons. The anatomical investigation of projections and the double-tracing with WGA and CTB showed that the pathways (i) and (ii), which would lead to high coherence values, are very weak, whereas the most prominent pathway (iii) (see figure below) leads, as reported and mentioned by the reviewer too, to low coherence both in CON and GE mice.

(a) Schematic of the retrograde CTB 555 injection in PFC and retro-/anterograde WGA injection in LEC. **(b)** Photograph depicting CTB-labeled and WGA-labeled neurons in the CA1 of a P10 CON mouse. **(c, d)** Photograph depicting the labeled CA1 neurons from the area marked by yellow and red boxes in (b) at higher-magnification.

The authors perform recordings in GE and control mice at two developmental periods: pre-juvenile (postnatal day 20-23, similar to the age in which the behavioral tests were performed) and neonate (postnatal day 8-10). In the pre-juvenile animals (sup. Fig. 3), they report no differences in LFP power in the three structures but firing rates are higher in the GE mice. In the neonatal GE mice (Fig. 2), LFP power is reduced in all three structures and LEC-HP as well as HP-PFC synchrony is reduced. Firing rates are not reported. These are two nice datasets but they are not presented in an ideal way and their divergent results are not discussed sufficiently. First, the pre-juvenile dataset is most relevant to the behavioral deficits since this is the age range at which the behavioral deficits were uncovered. I therefore think this supplementary figure deserves to be a main figure. Second, the authors should analyze the two datasets in exactly the same way so that they can be compared: synchrony should be quantified in the pre-juveniles and firing rates should be quantified in the neonates. Third, it is not clear why LFP power is reduced in neonatal GE mice but not in the older pre-juvenile GE mice. This is not discussed at all but of course requires an explanation: why would a deficit in early development disappear at a later developmental stage? This should be discussed after the two datasets have been analyzed in the same way in case additional differences between the results of the two datasets emerge.

In line with the recommendation of the reviewer, we illustrated the data from pre-juvenile mice in a main figure (Figure 2). Moreover, we quantified and displayed the firing rates for the neonatal mice and the synchrony for the pre-juvenile mice (Figure 2c, Figure 3b). In the revised manuscript, we discussed in more detail the why the early deficits are less prominent

at later developmental stage (Discussion, page 16, line 430-435).

Minor comments

Title: “Developmental decrease of entorhinal gate disrupts prefrontal-hippocampal communication” - as discussed above, a causal relationship between the entorhinal and the HPC-PFC deficits is not supported by the authors’ data. The title should be rewritten to reflect the main findings of the study, i.e. the LEC-HPC connectivity deficits.

We modified the title to better mirror the main findings of the study.

Line 51: “ ... the tasks ...” - not clear what kind of tasks is being referred to here

We specified the task.

The NOPd test is presented as a test of “associative” memory but I don’t see what association is being tested. In the testing phase, one of the two objects from the familiarization phase is replaced with a new one. The fact that the mice explore this object more simply shows that they recognize it as novel, not that they have formed any object-object association. It is of course possible that the animals form an association between the two objects in the familiarization phase and this shapes their exploratory behavior during the testing phase, but the current behavioral paradigm cannot distinguish this from the simpler non-associative novel object recognition). The OLP task, on the other hand, is associative. However, the deficit in the non-associative NOPd task in the GE mice suggests that the OLP deficit in these mice may reflect a more general deficit in object recognition.

In line with the available literature (Wilson, Langston et al. 2013; Chao, Huston et al. 2016), recognition of different objects in NOPd test requires the formation of an association between the two objects that is not present for two identical objects. To prove the distinct LEC contribution during a non-associative novel object recognition (NOR) task and NOPd, we chemogenetically silenced the LEC in the CON mice. For this, injection of AAV9_CaMKII_hM4Di_EGFP in the LEC of P1 CON mice was followed at P17 by i.p. injection DREADD agonist 21 (compound 21, C21) 45 min before NOR or NOPd. LEC silencing led to poorer performance in NOPd, but not NOR (see figure below).

(a) Schematic of the protocol for NOPd task and NOR task (top) and violin plots displaying the discrimination index in familiarization and test trials when averaged for CON and CON+C21 mice (bottom). (b) Same as (a) for NOR task. The black dotted line indicates chance level. Gray, blue and black dots correspond to investigated mice. Median and the 25th and 75th percentiles are displayed. *p<0.05, ***p<0.001.

While these new data demonstrate the contribution of LEC to associative recognition memory, we previously showed that GE mice have poor performance not only in this task but also in non-associative NOR, object-location recognition (OLR) and recency recognition (RR) tasks (Hartung, Cichon et al. 2016, Xu, Chini et al. 2019, Xu, Song et al. 2021). We discussed this issue in the revised manuscript (Results, page 5, line 137-140; page 6, line 151-153).

Line 142: “multiple - unit activity (MUA)” - is this multiple SUAs or “multi-unit” activity (i.e. unsorted spikes)?

Throughout the manuscript, we replaced the MUA (i.e. unsorted spikes) by SUA (i.e. sorted spikes) data.

Line 149: I see no reason to report logarithmic values of firing rates, this is unusual and makes the results less transparent. Also the base of the logarithm is not reported so it is not possible to convert these values to Hz. Please report all firing rates in spikes per second.

According to the suggestion of the reviewer, we reported the firing rates in spikes per second.

General: Indicate what error bars in figures represent and over what (animals? cells?) they are calculated.

We added the requested information.

Based on Figure 4, claims are made about the effectiveness of optogenetic LEC activation in driving LEC neurons (GE=CON) and HPC neurons (GE < CON). However, the authors only show single cell examples and no statistical tests are given to back up their statements. This would be an important result so the authors need to show the distribution of responses in both genotypes and perform the appropriate statistical tests.

We added the requested information.

Figure 5c: what does “Firing probability” mean? Also, the units are very different than in the similar Figure 6c. I also had the same question about the numbers mentioned when quantifying the efficiency of light stimulation (line 293).

To calculate firing probability for each single unit, we firstly defined ‘light-evoked spiking’ as spikes occurring in a 20 ms-long time window after the start of the light pulse. In a second step, the firing probability for a single unit was calculated as the number of light-evoked spikes divided by the total number of the pulses. In the revised manuscript, we added the information above to Methods section (page 26, line 690-691) and used the same unit for Fig. 5, 6 and 7.

The efficiency of light stimulation was used synonymously with the ‘firing probability’. To avoid confusions, we used throughout the manuscript only the term ‘firing probability’.

Line 310-311: The different ratios of excited cells in CON and GE mice - is this significant? Does the fraction of inhibited cells differ between the two groups?

We added to the manuscript the results of statistical comparison (page 14, line 326-327).

Chi-square test showed that the ratios of excited cells were significantly different in CON and GE mice ($p = 0.02$, $\chi = 5.43$). The ratios of the inhibited cells were comparable between the two groups ($p=0.77$, $\chi=0.082$).

Figure 5a: show at higher magnification so that the shape of individual neurons can be visualized

We followed the reviewer's suggestion and augmented the magnification in Fig. 6a (former Fig. 5a).

Figure 6 and Figure 5 present the results of the same experiment done in different parts of the circuit (LEC-PFC vs LEC-HPC), but the figures differ in their presentation. Please make them identical to facilitate comparison between these two experiments.

We modified Fig. 6 (former Fig. 5) and Fig. 7 (former Fig. 6) to facilitate the comparison between results.

In Figures 5 and 6, the responses are described as having an onset of 15 ms but I think the authors are referring to the latency of the maximum negativity of the response. It seems clear from the example traces that the onset of the response is much more rapid, as would be expected since it should only reflect the synaptic delay following terminal stimulation.

We added a brief definition of "onset" to Methods (page 21, line 560-561).

Figure 5c and 6c: are these individual neuron examples or averages over multiple neurons?

The plots display individual neuron examples. In the revised manuscript, we added the violin plots displaying the firing probability for each single unit in Fig. 6c (previous Fig. 5c) and Fig. 7e (former Fig. 6c).

In Figure 5d and 6d and 6e raster plots of neuronal responses to light stimulation are shown. It seems that each line of the raster plot is the response of one neuron. However, raster plots are usually shown for one neuron over multiple trials. It is difficult to make sense of a raster plot where each row is a different neuron (presumably also measured on different trials). I would therefore suggest the following to improve the presentation of the results. First, show a raster plot of at least one example neuron (as, for example, in the author's previous study Chini et al. 2020, Fig. 2). Second, show the average response of each neuron to the light stimulus, sorted by their response magnitude and direction (this seems to have been done already for the raster plots), as a color plot.

We modified the panels (currently Fig. 6c and Fig. 7e) in line with the reviewer's suggestions.

Figure 6d and e, right. It is difficult to see some of the data points. Make them larger. It is also not clear why the points have different sizes; if this is for indicating significance it is not necessary since this is also indicated by the y-axis.

We modified the Fig. 7f, g (former Fig. 6d, e) to improve the visibility and specified in the figure legend that the size of bubbles mirrors the firing rate of the single units.

Figure 7b: Are the graphs on the left examples of individual neurons? Please make clear. In these graphs, the maximum firing rate reaches ~ 1.5 whereas the graphs on the right report much stronger effects, 1.9 and 1.7.

Throughout the manuscript, we replaced MUA by SUA. Correspondingly, the Fig. 8b (former Fig. 7b) was modified to show the average SUA activity and the legend was amended to include this information. Moreover, the firing rate was plotted as the average spiking during the last two seconds of stimulation normalized to baseline spiking.

Line 321: “relative change” - does this mean the percentage change?

Since the “relative change” was calculated as fraction of values “during-pre” divided by “pre”, it has no unit.

Line 332: should be “hypothesized to occur during development” or something similar.

We rephrased the sentence.

Line 424: the neonatal animals are said to have been recorded in a “non-anesthetized state” - what does this mean exactly? Are the animals awake? Then the state should be described as such.

All mice have been recorded in the absence of anesthesia. However, since young mice spent a substantial time of time sleeping and the sleeping rhythms at this age lack the adult characteristics, we did not distinguish between the states. Therefore, we generally defined the state as “non-anesthetized”.

Line 442: how were electrodes inserted horizontally into the LEC? This would seem to require removing a considerable amount of tissue on the side of the head. It is difficult to see how this can be achieved without anesthesia

The term “horizontal” referred to the position of electrodes in relation to the plane of the animal (see figure below). To better describe the procedure, we rephrased to “electrodes were inserted vertically into LEC by placing them parallel to pups’ plane”.

(a) The hole (yellow dot) drilled for the LEC electrode is 6 mm to the right of the lambda, between the two veins (caudal rhinal vein and transverse sinus) (Mancini et. al, 2015). (b) Photograph of the pup during recordings with three electrodes inserted in PFC, HP, and LEC. (c) Nissl-stained coronal section showing the location of electrodes in the LEC.

Line 355: “giving” should be “given”

We corrected.

Line 363: "Under normal conditions, the neonatal LEC acts as gatekeeper of HP and PFC". It is not clear what the authors mean by 'gatekeeper', they should define precisely what they mean in terms of physiology.

We removed the misleading sentence.

Line 387-8: "Even if the passive and active membrane properties of entorhinal HP-projecting neurons were largely unaffected in GE mice, the neuronal function was impaired." - please explain what is meant by "neuronal function". As far as I can understand they have reported that oscillations in LEC are impaired (Figure 2A) and that they form weaker connections with the HP.

We rephrased this sentence.

References:

Mancini, M., A. Greco, E. Tedeschi, G. Palma, M. Ragucci, M. G. Bruzzone, A. R. D. Coda, E. Torino, A. Scotti, I. Zucca and M. Salvatore (2015). "Head and Neck Veins of the Mouse. A Magnetic Resonance, Micro Computed Tomography and High Frequency Color Doppler Ultrasound Study." PLOS ONE 10(6): e0129912.

Hartung H, et al. From Shortage to Surge: A Developmental Switch in Hippocampal–Prefrontal Coupling in a Gene–Environment Model of Neuropsychiatric Disorders. Cereb Cortex 26, 4265–4281 (2016).

Xu X, Chini M, Bitzenhofer SH, Hanganu-Opatz IL. Transient Knock-Down of Prefrontal DISC1 in Immune-Challenged Mice Causes Abnormal Long-Range Coupling and Cognitive Dysfunction throughout Development. J Neurosci 39, 1222 (2019).

Xu X, Song L, Hanganu-Opatz IL. Knock-Down of Hippocampal DISC1 in Immune-Challenged Mice Impairs the Prefrontal–Hippocampal Coupling and the Cognitive Performance Throughout Development. Cereb Cortex 31, 1240–1258 (2021).

Chao, O. Y., J. P. Huston, J. S. Li, A. L. Wang and M. A. de Souza Silva (2016). "The medial prefrontal cortex-lateral entorhinal cortex circuit is essential for episodic-like memory and associative object-recognition." Hippocampus 26(5): 633–645.

Wilson, D. I., R. F. Langston, M. I. Schlesiger, M. Wagner, S. Watanabe and J. A. Ainge (2013). "Lateral entorhinal cortex is critical for novel object-context recognition." Hippocampus 23(5): 352–366.

Reviewer #3

The article by Xu et al. details a series of experiments that examine LEC-HP-PFC dysfunction as a result of combined genetic and environmental factors in early development (GE). This GE model of psychiatric disorders is examined using sophisticated genetic tools that allow the authors to provide a very detailed account of how the LEC-HP-PFC network is affected in this model in terms of anatomy, electrophysiology and behavior. In general, the article provides compelling evidence that LEC- CA1 projections are compromised in the model where as LEC-PFC projections are much less affected. In general I think this is an interesting article but there are a number of issues that the authors need to address.

We thank the reviewer for the constructive feedback and most helpful comments and suggestions.

1. The rationale of the studies needs to be more precise. The GE model is cited at the end of the abstract as a 'model of disease.' This needs to be much more specific. The introduction continues in this vein by saying that the major burden of major psychiatric disorders is lifelong cognitive disability. This is also very non-specific. The introduction then becomes more focused on schizophrenia which makes sense so I would recommend that the article is reframed specifically in terms of schizophrenia (rather than 'disease') and the associated cognitive deficits and underlying neural mechanisms.

We better phrased the rationale of the study and highlighted the relevance of investigations in GE mice that represent a model of psychiatric risk mediated by gene-environment interactions. Due to the fact that neither Disc1 nor poly I:C are accepted as definitive risk factors for schizophrenia and have been related to multiple psychiatric disorders, we refrained for specifically relate the present results to this particular disease.

2. Pseudo-replication. There are a number of places in the manuscript where data are reported at the level of the slice rather than the animal. Worryingly, the figures suggest that analyses have also been carried out using slices as independent subjects. This will falsely increase statistical power. This is the case in the Fos study (Supp Fig 1) and also figures 3b/c and supp 5. These figures and analyses need to be replaced with analyses at the level of the animal.

In the revised manuscript, all analyses have been performed at animal level and, where necessary, the figures have been modified correspondingly (Supp Fig 1, Fig 3 and Supp Fig 5).

3. This issue is compounded by the fact that no information is given about sampling in these anatomical studies. Is every section counted? If so, how do the authors guard against counting the same cells in multiple sections? There is also no information given about how sections within regions are chosen. Given that different regions of LEC/HPC/PFC have different anatomical connectivity it is vital that the authors detail how they ensure they are consistently assessing the same regions within each area across mice.

We detailed the procedure to select slices and regions for quantification to the manuscript (Method, page 24, line 653-656).

4. There are some inconsistencies between numbers of animals reported, those presented in figures and those used for analysis. As an example, lines 106 says that 20 CON and 16 GE mice were used for the behavioral studies. However, Figure 1a has between 22-24 individual dots (hard to see as they sometimes overlap but a minimum of 22) for the control animals. The relevant ANOVA has degrees of freedom of (1,38) which shows that 40 animals were included in the analysis. None of these match up. The authors need to carefully check numbers cited for methods, figures and analyses throughout the manuscript.

Throughout the manuscript, we corrected the inconsistent values.

5. The first result reported is that young GE mice have poorer associative recognition memory. While the results of the OLP are consistent with this the NOPd are not as this is not a test of associative recognition. Mice can solve this based entirely on the fact that a novel object is presented. A simple object novelty signal (without the need to remember the association of 2 objects) could support the behaviour of the control mice.

In line with the available literature (Wilson, Langston et al. 2013; Chao, Huston et al. 2016), recognition of different objects in NOPd test requires the formation of an association between the two objects that is not present for two identical objects. To prove the distinct LEC contribution during a non-associative novel object recognition (NOR) task and NOPd, we chemogenetically silenced the LEC in the CON mice. For this, injection of AAV9_CaMKII_hM4Di_EGFP in the LEC of P1 CON mice was followed at P17 by i.p. injection DREADD agonist 21 (compound 21, C21) 45 min before NOR or NOPd. LEC silencing led to poorer performance in NOPd, but not NOR (see figure below).

(a) Schematic of the protocol for NOPd task and NOR task (top) and violin plots displaying the discrimination index in familiarization and test trials when averaged for CON and CON+C21 mice (bottom). (b) Same as (a) for NOR task. The black dotted line indicates chance level. Gray, blue and black dots correspond to investigated mice. Median and the 25th and 75th percentiles are displayed.

We added the new data (Results, page 5, line 115-120) and illustrated them in a new figure (supplementary Fig.2).

6. The discrimination index is defined as (time at object 2 - time at object 1) / (time at object

1 + time at object 2). This suggests that the identity of the novel object was not counterbalanced and object 2 was always the novel object. If this is the case how do the authors control for different levels of motivation to explore different objects? All details of counterbalancing need to be supplied.

To avoid misunderstanding, we changed the definition of the discrimination index to

NOPd: $(\text{time at novel object} - \text{time at old object}) / \text{time at both objects}$

OLP: $(\text{time at displaced object} - \text{time at stationary object}) / \text{time at both objects}$

in the revised manuscript (Method, page 22, line 603-606). This definition better reflects that the identity and position of the novel object was counterbalanced. Object 2 was not always in the same position, nor it always corresponds to the new object.

7. Total levels of exploration in both familiarization and test trials need to be presented and analyzed to ensure that the data are not affected by differential encoding or differential motivation to explore novelty.

As requested by the reviewer, we analyzed the total levels of exploration in both familiarization and test trials and added the data to the manuscript (Results, page 5, line 124-134; page 6, 143-145, Fig. 1 and supplementary fig. 3).

8. Was there a minimum level of exploration required in familiarization and test phases? Low exploration in familiarization will lead to poor encoding and discrimination indices in the test trial can be skewed by low levels of exploration. It is standard to have a minimum level of exploration for each trial.

In line with the reviewer's suggestion, we set a minimum level of exploration (20 cm/min) to exclude the animals with a low level of exploration. We modified the corresponding values (Results, page 5, line 126-136) and figure panels (Fig. 1).

9. How accurate is the tracking? How many frames are missed and is there a level at which trials are disregarded due to poor tracking?

We quantified the tracking accuracy for each mouse during familiarization and test trials of both NOPd and OLP tasks. The averaged tracking accuracy is 99.5 %. Representative tracking during one CON mouse performing NOPd tasks can be seen from the attached video. In the revised manuscript, we ignored the mice with more than 5% missing frames.

10. How do the findings in Figure 3 relate to the Layer 2a and 2b distinction described by Leitner et al. 2016 (Nat Neurosci) and Vandrey et al. 2020 (Current Biology).

Leitner et al. identified two distinct populations of excitatory neurons in layer 2 of LEC: reelin-positive neurons confined to the more superficial part 2a calbindin-positive neurons located in layer 2b. Both studies mentioned by the reviewer also showed that the projections of these two populations are segregated, with the reelin-positive neurons projecting to hippocampal DG area. To relate our results to the studies above, we assessed the histochemical identity of entorhinal neurons by injecting the retrograde tracer FG in CA1 or PFC of P7 mice and

subsequently performing calbindin and reelin staining. Our results show that CA1-projecting neurons are calbindin-positive, whereas the PFC-projecting neurons are reelin-positive (see figure below, a, b). Thus, layer 2a neurons not only project to DG but also to PFC.

To investigate whether the same Layer 2a reelin-positive LEC neurons project to both DG and PFC, we injected the retrograde tracer CTB 555 in DG and CTB 488 in PFC of a P7 mouse. The mouse was perfused 3 days after the injection. There was almost no overlap between the PFC-projecting and DG-projecting LEC neurons (see figure below, c). These results show that during development, PFC-projecting and DG-projecting LEC neurons are both reelin-positive and located in Layer 2a of LEC.

We briefly related these results to the literature data mentioned by the reviewer (Results, page 9, line 240-243).

(a) Schematic of the retrograde FG injection in HP. Middle, photograph depicting FG-labeled neurons in the LEC of a P10 CON mouse. Right, FG and calbindin-labeled neurons in LEC. (b) Schematic of the retrograde FG injection in PFC. Middle, photograph depicting FG-labeled neurons in the LEC of a P10 CON mouse. Right, FG and reelin-labeled neurons in LEC. (c) Schematic of the retrograde CTB 488 injection in PFC, CTB 555 injection in DG. Middle, photograph depicting CTB 555-labeled neurons in the LEC of a P10 CON mouse. Right, CTB 488-labeled neurons and CTB 555-labeled neurons in the LEC.

References:

Chao, O. Y., J. P. Huston, J. S. Li, A. L. Wang and M. A. de Souza Silva (2016). "The medial prefrontal cortex-lateral entorhinal cortex circuit is essential for episodic-like memory and associative object-recognition." *Hippocampus* 26(5): 633-645.

Wilson, D. I., R. F. Langston, M. I. Schlesiger, M. Wagner, S. Watanabe and J. A. Ainge (2013). "Lateral entorhinal cortex is critical for novel object-context recognition." *Hippocampus* 23(5): 352-366.

REVIEWER COMMENTS

Reviewer #1 (Remarks to the Author):

The authors have elegantly addressed most of my concerns. The current version of the manuscript is much improved and very interesting.

I have only minor comments but I really think these should be addressed to improve the manuscript quality to the level it deserves.

-The main issue is with the way the manuscript is written. The abstract, particularly is very poorly written. Someone who did not read the paper will not understand what has been done. The third sentence (Here we show that the... psychiatric risk) does not make sense. We do not know what GE means, nor what you mean by "the combined genetic and environmental etiology of psychiatric risk". Similarly, what is a "disease-characteristic cognitive disability". It may be helpful to consult with editing companies so that readers can really understand what has been performed in this study.

p3, line 69: what do you mean by "tackled"?

There are issues with a lot of the references used in the manuscript:

-Reference 10 is about mesoamerican ceramics... I don't see what it has to do with the current study.

-p3 line 71: reference 14 is about mRNA expression and phospholemman, a phosphoprotein involved in chloride channel expression and regulation. reference 15 is about pre-alpha cells in the entorhinal cortex of schizophrenia patients. Could you provide references on "cellular and synaptic deficits and aberrant axonal innervation"?

-p3, lines 72-73: can you provide references supporting this statement?

-p3, line 75. please rephrase this sentence, references 16, 17 (Bolkan et al 2018) were not done in a schizophrenia model. you may mean that alterations of thalamo-cortical communication (what you call broader network dysconnectivity) are responsible for cognitive deficits observed in schizophrenia (refs 16-17) and are observed in animal models (ref 18).

-on Figure 9, it is not clear what you call "relative power". What is it relative to? Why is it below zero even before stimulation?

-It still would be important to provide direct evidence (and not simulated data) that stimulations do not cause lesions. Maybe Fluoro-jade? Can you show LFP data instead of units?

-The Kolmogorov-Smirnov test does not test for homogeneity of variance but more for equality of distributions. More appropriate tests would be an F-test, Bartlett's test or a Lavene's median test.

Reviewer #2 (Remarks to the Author):

The authors have addressed my concerns for the most part satisfactorily. They have also added new data and analyses to address the concerns of other reviewers. As a result, the manuscript is much improved and I am confident that it will make an important contribution to our understanding of LEC-HC-PFC network dysfunction in psychiatric disease. I have some additional minor comments and suggestions for improvement that should be addressed prior to publication.

Line 86: "limbic brain areas" is too vague, please specify which areas are meant here

Line 100: "On its turn" should be "In turn"

For the new results in supplementary figure 2, the authors should explain the statistical test on which their claims on the effects of LEC inactivation are based. In their rebuttal letter, the authors

present a statistical comparison between the LEC inactivation group and (it seems) the control group from the original manuscript. This would be helpful to also include in supplementary figure 2. One potential concern is that the hM4Di group is lacking the appropriate controls (i.e. mice injected with eGFP virus only and injected with C21), leaving open the possibility that the effects could reflect the virus injection itself and/or injection of the C21 agonist. Arguing against such non-specific effects, however, is the lack of effect of LEC inactivation on the novel object recognition (NOR) task as the authors demonstrate in their rebuttal letter. These results should therefore also be included in Supplementary Figure 2 in order to strengthen the author's conclusion.

I still am not entirely convinced that the NOPd task is 'associative' since, as reviewer 3 also pointed out, the mice can solve this task simply by recognizing the novel object and without forming any object-object association. However, the authors' new LEC inactivation results suggest that NOPd task involves more than just the recognition of a novel object since LEC inactivation did not impair performance on the NOR task. As mentioned above, the NOR results are only presented in the rebuttal letter but I think they should be included in the supplementary figure 2 to help convince readers that the NOPd task is not simply a form of novelty recognition.

The design of the NOPd and OLP tasks, their naming as well as the way they are interpreted, is based on the Chao et al. 2016 study which the author's cite in the rebuttal letter. However, this study is not cited in the manuscript but should of course be cited when the authors introduce these two tasks starting on line 111. The Wilson et al. 2013 study which is cited here is not appropriate since it does not include the NOPd task.

Figure 1: the x-axes could be made more informative than '1' and '2', it took me a while to understand what these mean. Better would be e.g. 'Object 1', 'Object 2', or 'Novel', 'Familiar'

The presentation of the results from the neonatal and pre-juvenile cohorts is much improved and it is now easier to compare the two datasets. Two minor comments: 1) I am not sure whether oscillatory power is quantified in exactly the same way, the y-axes in Figures 2a and 3a look like they have different units for power. 2) Can the authors also calculate imaginary coherence for the pre-juveniles?

Now that it is easier to directly compare the results from pre-juvenile and neonatal datasets, it is clear that activity in the LEC-HC-PFC circuit is quite different at these two developmental timepoints. I appreciate that the authors have now addressed these differences in the discussion. What they do not consider, however, is that the pre-juveniles are recorded under anesthesia whereas the neonates are not. Can the authors speculate whether these methodological differences could have contributed to the divergent results obtained in these two developmental cohorts?

Supplementary Figure 10: In a, the author's should adjust the color scale (i.e. maximum value) so that the temperature changes are easier to see. Also, in b,c and d it is unclear what 'units X trials' means. A raster plot should only show the activity of one neuron over trials.

Line 987: "single unit" - surely the authors here mean the plural "single units"

"Firing probability" still needs to be properly defined in the methods. In their rebuttal letter, the authors explain how this is calculated but I couldn't find this in the methods. 'Light-evoked spiking' is defined (lines 699-700) but this seems to be different from "firing probability"

The authors in their rebuttal letter have explained to me what 'non-anesthetized' means but this should also be explained in the manuscript since many readers will likely wonder about what this means (as I did). I think it is actually a major strength of the study that the recordings were carried out in non-anesthetized neonatal mice.

6e, f and 7f,g are described as 'raster plots' but aren't these the average responses of each neuron (i.e. PSTHs)? Please correct accordingly

Reviewer #3 (Remarks to the Author):

The authors have addressed my comments and I would recommend publication.

Reviewer #1:

The authors have elegantly addressed most of my concerns. The current version of the manuscript is much improved and very interesting.

We thank the reviewer for the feedback and helpful comments.

I have only minor comments but I really think these should be addressed to improve the manuscript quality to the level it deserves.

The main issue is with the way the manuscript is written. The abstract, particularly is very poorly written. someone who did not read the paper will not understand what has been done. The third sentence (Here we show that the.... psychiatric risk) does not make sense. we do not know what GE means, nor what you mean by “the combined genetic and environmental etiology of psychiatric risk”. Similarly, what is a “disease-characteristic cognitive disability”. It may be helpful to consult with editing companies so that readers can really understand what has been performed in this study.

We rephrased the abstract.

p3, line 69: what do you mean by “tackled”?

We replaced ‘tackled’ by ‘addressed’.

There are issues with a lot of the references used in the manuscript:

We apologize for the mistakes that resulted from EndNote bugs. In the revised version, all references have been manually checked.

-Reference 10 is about mesoamerican ceramics... I don't see what it has to do with the current study.

We removed the reference.

-p3 line 71: reference 14 is about mRNA expression and phospholemman, a phosphoprotein involved in chloride channel expression and regulation. reference 15 is about pre-alpha cells in the entorhinal cortex of schizophrenia patients. Could you provide references on “cellular and synaptic deficits and aberrant axonal innervation”?

We removed the wrong references and added the required references (page 3, line 71).

-p3, lines 72-73: can you provide references supporting this statement?

We added the required references.

-p3, line 75. please rephrase this sentence, references 16, 17 (Bolkan et al 2018) were not done in a schizophrenia model. you may mean that alterations of thalamo-cortical communication (what you call broader network dysconnectivity) are responsible for cognitive deficits observed in schizophrenia (refs 16-17) and are observed in animal models (ref 18).

We rephrased the sentence.

-on Figure 9, it is not clear what you call “relative power”. What is it relative to? Why is it below zero even before stimulation?

The ‘relative power’ was calculated as $(Power_{post} - Power_{pre}) / Power_{pre}$ for light-induced inhibition of entorhinal terminals in HP of CON mice and as $(Power_{GE} - Power_{CON}) / Power_{CON}$ for non-stimulated GE mice. By this means, the power changes related to the activity before the stimulation (for light-induced inhibition of entorhinal terminals in HP of CON mice) and to the activity of non-stimulated CON mice (for non-stimulated GE mice) have been monitored.

The negative values correspond to lower oscillatory power after stimulation when compared to the values before the stimulation. In the revised version, we replace “relative power” by “normalized power” and added a short definition to the figure legend. We applied same change to the term ‘relative coherence’ that was re-defined as ‘normalized coherence’.

-It still would be important to provide direct evidence (and not simulated data) that stimulations do not cause lesions. Maybe Fluoro-jade? Can you show LFP data instead of units?

In line with the reviewer’s concerns, we previously performed an in-depth analysis of the effects of optogenetic manipulation on developing circuits (Bitzenhofer et al., 2017 – Nature Commun 8:14563, Bitzenhofer et al., 2017 – Front Cell Neurosci 11:239; Ahlbeck et al., 2018 – eLife 7:e33158). We showed that acute light stimulation (3 ms-pulse, sinusoidal, ramp) did not affect the firing activity in the PFC of opsin-free mice. More recently, we developed a chronic light stimulation paradigm for neonatal mice (from postnatal day (P) 7 to 11, 473 nm wavelength, 3 s duration, 7 s interval, 180 repetitions, 30 min total duration) (Bitzenhofer et al. 2021, Neuron 109(8):1350-1364). Throughout development, the chronic stimulation did not affect the neuronal density (Fig. S3 in Bitzenhofer et al., 2021). To assess the neuronal death after the stimulation, we performed caspase3 staining. Very few caspase-positive neurons (green, 1-3 neurons/investigated area/slice) have been detected.

As requested, we performed additional analyses to monitor the LFP power in opsin-free mice. Due to the large light artifacts that mask the oscillatory activity (see also Bitzenhofer et al., 2017 – Nature Commun), the LFP power was calculated as relative value:

$$(Power_{post} - Power_{pre}) / Power_{pre},$$

where *pre* corresponds to the power before stimulation window and *post* to the power after the stimulation window. The LFP power in opsin-free mice was not affected by stimulation, indicating that the light stimulation does not harm the developing networks. We added the new data to the manuscript (page 11, lines 276-277) and displayed them in the Supplementary Figure 10.

-The kolmogorov smirnov test does not test for homogeneity of variance but more for equality of distributions. More appropriate tests would be an F-test, Bartlett’s test or a Lavene’s median test.

In line with the suggestion, we used the F-test to test for homogeneity of variance.

References:

Ahlbeck J, Song L, Chini M, Bitzenhofer SH, Hanganu-Opatz IL. Glutamatergic drive along the septo-temporal axis of hippocampus boosts prelimbic oscillations in the neonatal mouse. *eLife* 7, e33158 (2018).

Bitzenhofer SH, Ahlbeck J, Wolff A, Wiegert JS, Gee CE, Oertner TG, & Hanganu-Opatz IL. (2017). Layer-specific optogenetic activation of pyramidal neurons causes beta-gamma entrainment of neonatal networks. *Nat Commun*, 8, 14563.

Bitzenhofer SH, Ahlbeck J, Hanganu-Opatz IL. Methodological Approach for Optogenetic Manipulation of Neonatal Neuronal Networks. *Front Cell Neurosci* 11, 239 (2017).

Bitzenhofer SH, Pöplau JA, Chini M, Marquardt A, Hanganu-Opatz IL. A transient developmental increase in prefrontal activity alters network maturation and causes cognitive dysfunction in adult mice. *Neuron* 109, 1350-1364.e1356 (2021).

Reviewer #2:

The authors have addressed my concerns for the most part satisfactorily. They have also added new data and analyses to address the concerns of other reviewers. As a result, the manuscript is much improved and I am confident that it will make an important contribution to our understanding of LEC-HC-PFC network dysfunction in psychiatric disease. I have some additional minor comments and suggestions for improvement that should be addressed prior to publication.

We thank the reviewer for the feedback and helpful suggestions.

Line 86: "limbic brain areas" is too vague, please specify which areas are meant here

We replaced "limbic brain areas" by "HP and PFC".

Line 100: "On its turn" should be "In turn"

We modified.

For the new results in supplementary figure 2, the authors should explain the statistical test on which their claims on the effects of LEC inactivation are based. In their rebuttal letter, the authors present a statistical comparison between the LEC inactivation group and (it seems) the control group from the original manuscript. This would be helpful to also include in supplementary figure 2. One potential concern is that the hM4Di group is lacking the appropriate controls (i.e. mice injected with eGFP virus only and injected with C21), leaving open the possibility that the effects could reflect the virus injection itself and/or injection of the C21 agonist. Arguing against such non-specific effects, however, is the lack of effect of LEC inactivation on the novel object recognition (NOR) task as the authors demonstrate in their rebuttal letter. These results should therefore also be included in Supplementary Figure 2 in order to strengthen the author's conclusion.

As suggested, we specified that the statistical comparison has been done between the LEC inactivation groups and the control group (from the original manuscript). We added the corresponding plots to Supplementary Figure 2.

In line with the reviewer's suggestion, we added the results illustrating the effects of LEC inhibition on NOR performance (NOP in the revised manuscript) (page 5, line 118-119) and displayed them in Supplementary Figure 2.

I still am not entirely convinced that the NOPd task is 'associative' since, as reviewer 3 also pointed out, the mice can solve this task simply by recognizing the novel object and without forming any object-object association. However, the authors' new LEC inactivation results suggest that NOPd task involves more than just the recognition of a novel object since LEC inactivation did not impair performance on the NOR task. As mentioned above, the NOR results are only presented in the rebuttal letter but I think they should be included in the supplementary figure 2 to help convince readers that the NOPd task is not simply a form of novelty recognition.

We added the results to the revised manuscript (page 5, line 118-119) and Supplementary Figure 2.

The design of the NOPd and OLP tasks, their naming as well as the way they are interpreted, is based on the Chao et al. 2016 study which the author's cite in the rebuttal letter. However, this study is not cited in the manuscript but should of course be cited when the authors introduce these two tasks starting on line 111. The Wilson et al. 2013 study which is cited here is not appropriate since it does not include the NOPd task.

We added the missing reference.

Figure 1: the x-axes could be made more informative than '1' and '2', it took me a while to understand what these mean. Better would be e.g. 'Object 1', 'Object 2', or 'Novel', 'Familiar'
We modified the description of x-axes in Figure 1.

The presentation of the results from the neonatal and pre-juvenile cohorts is much improved and it is now easier to compare the two datasets. Two minor comments: 1) I am not sure whether oscillatory power is quantified in exactly the same way, the y-axes in Figures 2a and 3a look like they have different units for power. 2) Can the authors also calculate imaginary coherence for the pre-juveniles?

As noticed by the reviewer, the different units for power in neonatal and pre-juvenile mice result from different calculation procedure applied to oscillatory activity with distinct properties. At neonatal age, the network activity is discontinuous and therefore, the oscillatory power was calculated as averaged power spectra of the oscillatory episodes normalized to the baseline power of time windows lacking oscillatory activity. At prejuvenile age, continuous network oscillations were recorded. Thus, the oscillatory power was calculated for the entire recording. We specified the different methods to calculate the power in Materials & Methods (page 26, lines 689-692).

The synchrony within prefrontal-hippocampal-entorhinal network and its underlying mechanisms are the topic of a separate study. The preliminary investigations revealed that at pre-juvenile age an apparent hyper-coupling links the three brain areas. This "hyper-coupling" relates to the age-dependent embedding of interneurons into circuits that differently occurs in controls and GE mice. As a result, while the coherence in controls decreases from neonatal to pre-juvenile stage and augments afterwards until adulthood, it remains stable in GE mice during the entire postnatal development. Consequently, at pre-juvenile age a "hyper-coupling" was detected in GE mice vs. controls. To avoid confusions, we decided to not include the synchrony analysis in the present manuscript but to investigate the topic in more detail in a separate study.

Now that it is easier to directly compare the results from pre-juvenile and neonatal datasets, it is clear that activity in the LEC-HC-PFC circuit is quite different at these two developmental timepoints. I appreciate that the authors have now addressed these differences in the discussion. What they do not consider, however, is that the pre-juveniles are recorded under anesthesia whereas the neonates are not. Can the authors speculate whether these methodological differences could have contributed to the divergent results obtained in these two developmental cohorts?

We added a brief discussion of the topic to the manuscript (page 16, line 432-435).

Supplementary Figure 10: In a, the author's should adjust the color scale (i.e. maximum value) so that the temperature changes are easier to see. Also, in b,c and d it is unclear what 'units X trials' means. A raster plot should only show the activity of one neuron over trials.

As requested, we modified the figure.

Line 987: "single unit" - surely the authors here mean the plural "single units"

We corrected.

"Firing probability" still needs to be properly defined in the methods. In their rebuttal letter, the authors explain how this is calculated but I couldn't find this in the methods. 'Light-evoked spiking' is defined (lines 699-700) but this seems to be different from "firing probability"

We added the definition to Materials and Methods section (page 26, line 697-700).

The authors in their rebuttal letter have explained to me what 'non-anesthetized' means but this should also be explained in the manuscript since many readers will likely wonder about what this means (as I did). I think it is actually a major strength of the study that the recordings were carried out in non-anesthetized neonatal mice.

We specified the term in Materials and Methods section (page 18, line 479-482).

Fig6e, f and 7f,g are described as 'raster plots' but aren't these the average responses of each neuron (i.e PSTHs)? Please correct accordingly

We modified the legend correspondingly.

Reviewer #3:

The authors have addressed my comments and I would recommend publication.

We thank the reviewer for the feedback.